# *SETBP1* variants outside the degron disrupt DNA-binding, transcription and neuronal differentiation capacity to cause a heterogeneous neurodevelopmental disorder

Different types of germline de novo *SETBP1* variants cause clinically distinct and heterogeneous neurodevelopmental disorders: Schinzel-Giedion syndrome (SGS, via missense variants at a critical degron region) and *SETBP1*-haploinsufficiency disorder. However, due to the lack of systematic investigation of genotype-phenotype associations of different types of *SETBP1* variants, and limited understanding of its roles in neurodevelopment, the extent of clinical heterogeneity and how this relates to underlying pathophysiological mechanisms remains elusive. This imposes challenges for diagnosis. Here, we present a comprehensive investigation of the largest cohort to date of individuals carrying *SETBP1* missense variants outside the degron region ($n = 18$). We performed thorough clinical and speech phenotyping with functional follow-up using cellular assays and transcriptomics. Our findings suggest that such variants cause a clinically and functionally variable developmental syndrome, showing only partial overlaps with classical SGS and *SETBP1*-haploinsufficiency disorder. We provide evidence of loss-of-function pathophysiological mechanisms impairing ubiquitination, DNA-binding, transcription, and neuronal differentiation capacity and morphologies. In contrast to SGS and *SETBP1* haploinsufficiency, these effects are independent of protein abundance. Overall, our study provides important novel insights into diagnosis, patient care, and aetiology of SETBP1-related disorders.

Different types of germline variants in the gene encoding SET binding protein 1 (*SETBP1*) (NM_015559.2) lead to distinct and phenotypically heterogeneous developmental syndromes[1]. Schinzel–Giedion syndrome (SGS, MIM #269150)[2] is a rare and severe disorder with multi-system malformations that include recognisable facial characteristics, neurological problems (including severe intellectual disability, intractable epilepsy, cerebral blindness and deafness) and various congenital anomalies (such as heart defects, and genital, kidney and bone abnormalities)[3–6]. The majority of patients with a molecularly confirmed diagnosis have a shortened lifespan and often do not survive beyond the first decade[6]. In 2010, heterozygous germline de novo missense variants in one section of the *SETBP1* gene were identified as a cause of SGS. The variants cluster in a hotspot of 12 base pairs coding for four amino acids (residues 868–871) in the SKI domain of the

✉e-mail: maggie.wong@mpi.nl; simon.fisher@mpi.nl

SETBP1 protein[2,6]. These four amino acid residues are part of a canonical degron sequence recognised by ubiquitin E3 ligases and important for regulating protein degradation, and the variants are thought to cause SGS via gain-of-function mechanisms. Intriguingly, overlapping somatic *SETBP1* variants have been identified recurrently in several forms of myeloid leukaemia[6,7]. These overlapping somatic variants are more disruptive to the degron, implying a higher functional threshold to cause cancer[6], and it is thought that germline variants might lead to features similar to cancer cells, such as increased proliferation and accumulation in DNA damage, but via different cell type-specific pathways[8].

In contrast, *SETBP1*-specific deletions and de novo loss-of-function (LoF) point mutations disrupting the gene result in *SETBP1*-haploinsufficiency disorder[9], (MIM #616078), a syndrome that is phenotypically milder than SGS[10–14]. The heterozygous *SETBP1*-specific deletions lead to a loss of the locus, while the LoF variants (nonsense and frameshift variants) are predicted to lead to nonsense-mediated decay of the mutated *SETBP1* transcript, both of which therefore result in reduced dosage of SETBP1 protein (haploinsufficiency). Unlike classical SGS, individuals carrying such variants do not show major congenital or growth anomalies. Our recent systematic gene-driven studies of a large cohort with confirmed *SETBP1* LoF variants revealed a far broader clinical severity spectrum than previously reported[15,16]. Despite subtle overlapping facial dysmorphisms, and in contrast to SGS, the affected individuals do not present with a recognisable facial gestalt or specific features of dysmorphisms. The main clinical features include moderate-to-severe speech and language impairments, mild motor developmental delay, wide variability in intellectual functioning, hypotonia, vision impairment, and behavioural problems such as attention/concentration deficits and hyperactivity[15,16]. Heterozygous pathogenic LoF variants in *SETBP1* have been independently identified by exome/genome sequencing in different cohorts of individuals with childhood apraxia of speech (CAS)[13,17,18]. In a comprehensive speech phenotyping study led by Morgan et al.[15], poor communication was shown to be a central feature of *SETBP1* haploinsufficiency disorder. These findings further highlight the heterogeneity and complexity of SETBP1-related aetiologies.

*SETBP1* is expressed in numerous tissues[19], including the brain, and multiple alternative transcripts encoding different isoforms have been found [OMIM 611060] (Supplementary Fig. 1a). However, there is little existing knowledge concerning the expression and functions of the SETBP1 protein, as well as its isoforms, with much of our understanding coming from studies of somatic variants in haematopoietic cells or overexpression systems. Accumulation of SETBP1 has been shown to reduce protease cleavage of its interactor oncoprotein SET, resulting in the formation of a SETBP1-SET-PP2A complex that inhibits PP2A phosphatase activity, thus promoting proliferation of leukaemic cells[6,7,20,21]. SETBP1 has chromatin remodeller functions, binding to AT-rich genomic regions via two AT-hooks, and has been shown to recruit an HCF1/KMT2A/PHF8 epigenetic complex in HEK cells overexpressing SETBP1[22]. Acting as a transcription factor, SETBP1 is able to activate or repress expression of genes such as *HOXA9, HOXA10, RUNX1, MYB* and *MECOM* in haematopoietic cells and in HEK cells overexpressing an SGS variant[22–25]. Nevertheless, the roles of SETBP1 in the developing brain and the pathophysiological pathways underlying germline pathogenic *SETBP1* variants remain elusive. Thus far, five studies have investigated the molecular consequences of *SETBP1* disruptions in mouse and human neuronal models[8,22,26–28], all of which described impairment in cell proliferation as a shared mechanism. Overexpression of human wild-type SETBP1 and an SGS variant in mouse embryos via *in utero* electroporation disrupted neuronal migration and neurogenesis in the neocortex[22]. In neural progenitor cells derived from patients with SGS, accumulation of SETBP1 promoted cell proliferation and induced DNA damage via SET stabilisation that inhibited P53, and a PARP-1-dependent mechanism[8]. Investigations of neural

progenitors in which *SETBP1* had been knocked out identified prolonged proliferation and distorted layer-specific neuronal differentiation with an overall decrease in neurogenesis via the WNT/β-catenin signalling pathway[26]. However, these studies focused on a few classical and atypical SGS variants, two LoF variants, or *SETBP1* knockouts, leaving the majority of missense variants uncharacterised.

In 2017, Acuna-Hidalgo et al. identified four individuals carrying *SETBP1* variants in close proximity to the canonical degron [p.(Glu862Lys), p.(Ser867Arg), p.(Thr873Ile)] who showed a milder developmental phenotype with clinical characteristics that partially overlapped with classical SGS[6]. Additional individuals carrying missense variants in close proximity to the degron have since been reported in the medical literature. Nevertheless, the majority of the variants reported thus far are classified as variants of uncertain significance, and their functional impacts have not been thoroughly characterised. Genotype–phenotype associations of germline *SETBP1* variants, therefore, remain unclear. The highly variable phenotypes and severity seen in these patients have made diagnoses difficult and precluded the development of new, personalised therapies, often creating confusion among clinicians and patient families.

In this work, we address these important issues using a gene-driven approach. We recruit the largest cohort of individuals, to our knowledge, with a molecular diagnosis of *SETBP1* variants (missense variants and one in-frame deletion) outside the degron region (i.e. not affecting amino acids 868-871) who show clinical features that do not fit with the original SGS diagnostic criteria. We investigate the genotype–phenotype relationships of variants in the *SETBP1* gene by thorough phenotyping of clinical and speech/language features, as well as functional follow-up of specific variants in cellular assays. We discover that *SETBP1* variants outside the degron lead to a neurodevelopmental disorder characterised by a broad spectrum of clinical features, which are much milder than classical SGS despite partial overlaps, but in some cases more severe than *SETBP1*-haploinsufficiency disorder. Using functional cellular assays, we show that the effects of SETBP1 variants involve a combination of mechanisms, including deficits in protein degradation, ubiquitination and transcriptional control, independent of elevated protein levels. Using transcriptomics, we demonstrate that patient fibroblasts carrying three classes of *SETBP1* variants show transcriptomic profiles, with only partial overlaps, affecting biological processes related to distinct gene ontologies. Finally, we investigate the functional impacts of these different groups of variants in patient cell-derived induced neurons. We show that the differentiation capacity, neuron morphologies, and transcriptomic profiles of all variants tested are affected with differences between and within groups. Overall, our findings point towards pathophysiological mechanisms that act via functional dosage of SETBP1 protein in a way that is dependent on variant location and specific change in amino-acid residues, rather than merely the amount of protein, explaining the clinical heterogeneity observed in patients.

## Results
### SETBP1 variants outside the canonical degron of the SETBP1 protein cluster in the SKI domain
In our study, we included 18 unrelated individuals carrying rare heterozygous variants with uncertain functional impact in *SETBP1* (NM_015559.2), a gene under constraint against LoF and missense variation [pLoF: $o/e = 0.02$ (0.01–0.11); missense: $o/e = 0.9$ (0.84–0.95); gnomAD v.2.1.1[29] (Fig. 1a and Supplementary Data 1). Variants were identified via diagnostic exome or genome sequencing in various diagnostic genetic laboratories in different countries. In one case, the variant was first identified with direct Sanger sequencing of the *SETBP1* gene based on clinical observations, followed by trio-based exome sequencing. We collected the clinical and genotype information from the individuals. None of the 18 individuals met the diagnostic criteria of Schinzel–Giedion Syndrome (SGS)[30]. Fifteen

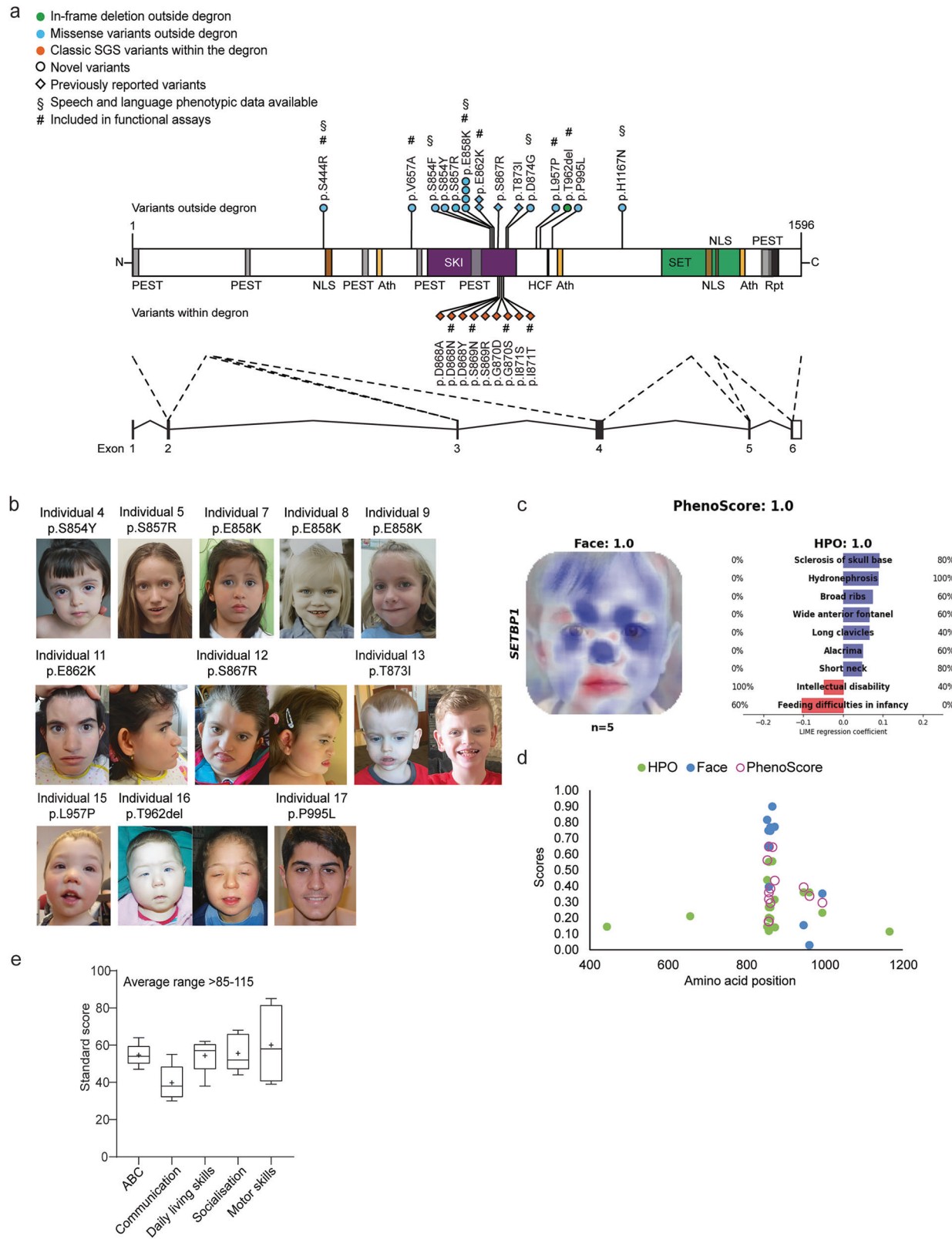

individuals carried a de novo *SETBP1* variant, while the other two had inherited a variant from an affected parent. One individual harboured a missense variant that was not inherited from the mother; the father was unavailable for testing. Among the 15 individuals with a de novo variant, 14 carried a missense variant and one had an in-frame deletion [c.2885_2887del(CCA) p.(Thr962del)]. Within our cohort, there were multiple cases of recurrent identical de novo

variants, including c.2572 G > A p.(Glu858Lys) found in four children and c.2584 G > A p.(Glu862Lys) in two individuals, revealing an independent mutational hotspot located in close proximity to the canonical degron region. None of the *SETBP1* variants included in our study were present in the gnomAD v.2.1.1 database. Two individuals also carried variants affecting other known disease genes. In proband 3, who has a de novo c.2561 C > T p.(Ser854Phe) *SETBP1* variant, an

**Fig. 1 | SETBP1 variants outside the degron cluster in the SKI domain, facial photographs and speech phenotyping of individuals with SETBP1 variants.**
**a** Schematic representation of the SETBP1 protein (UniProt: Q9Y6X0) indicating the locations of variants included in this study. These comprise eleven novel germline variants (circles), including ten missense variants (blue) and one in-frame deletion (de novo, green) outside the canonical degron. Three missense variants outside (blue) and nine within (orange) the degron that were previously reported (diamond) are also annotated in the schematic. § represent variants for which speech phenotyping data were available. # represent variants included in functional assays. Five exons (black bars) encode isoform A of the protein (NP_056374.2, 1596 amino acids). The known SETBP1 protein domains are indicated[7,21,22]. An overview with variant details per subject is provided in Supplementary Data 1. **b** For individuals with an amino acid change in the 854–858 region, no visually recognisable facial features could be delineated. Shared facial features present in at least three out of five individuals with an amino acid change in 862–874 include prominent ears, shallow orbits, midface retraction, and microcephaly. Individuals with a 962 and 957 variant show similar facial features, including a round face, blepharophimosis, ptosis, hypertelorism, and a short nose with a bulbous tip, features that are also often noted in individuals with SETBP1 haploinsufficiency disorder. **c** PhenoScore is able to distinguish individuals with SGS and SETBP1-haploinsufficiency disorder. Subgroup analysis of (left) facial photographs and (right) phenotypic data (HPO terms) of five individuals with SGS and five individuals with SETBP1-haploinsufficiency disorder. Score: 0 = SETBP1-haploinsufficiency disorder; 1 = SGS.
**d** Phenotypic similarity of individuals with missense SETBP1 variants outside the degron predicted using PhenoScore (purple circle), HPO terms only (green) or facial features only (blue). Score: 0 = SETBP1-haploinsufficiency disorder; 1 = SGS. Each datapoint represents one individual (n = 18). Details in Supplementary Data 2. **e** Performance on Vineland-3 subtests. Lines denote median scores; X denotes mean scores; ABC, Adaptive Behaviour Composite. Standard scores between 85 and 115 are considered within the average range, with a mean of 100 and a standard deviation of 15. The performance of five individuals was measured. All details of statistical tests and p-values are provided in Source Data 2. Source data are provided as a Source Data file.

EHMT1 variant was identified. In proband 1, who has a c.1332 C > G p.(Ser444Arg) SETBP1 variant, an inherited 5q22.31 dup and WWOX variants were identified. However, these additional variants in probands 1 and 3 are unlikely to be pathogenic based on the patients' clinical features, the inheritance model, and the lack of functional impact observed in assays performed in cellular models (unpublished observations by co-author M.S.H. with the EHMT1 and WWOX variants).

## SETBP1 variants outside the canonical degron cause a broad spectrum of clinical features

An overview of the main clinical features per individual is shown in Supplementary Data 1. Individuals (n = 5) with a variant in close proximity to the degron within the SKI domain (affecting amino acids 862–874, excluding the degron 868–871) showed severe or profound intellectual disability and severe motor impairment with inability to walk without support. Four of these individuals were unable to speak. Three individuals showed spasticity. Focal and tonic-clonic seizures were noted in two of these individuals. Two individuals showed renal abnormalities: one had mild kidney dilatation, and another had medullary cystic kidneys. Shared facial features were present in at least three out of the five individuals, including prominent ears, shallow orbits, midface retrusion and microcephaly (Fig. 1b). Overall, the phenotypes of these five children did not fulfil the original Lehman et al. criteria for SGS[30]. However, based on the severity of the phenotypes and facial features, they appeared more similar to (but less severe than) SGS compared to SETBP1-haploinsufficiency disorder.

Individuals (n = 7) with variants located slightly further away from the degron but still within the SKI domain (amino acids 854–858) showed mild or moderate intellectual disability. Two out of four individuals with a c.2572 G > A p.(Glu858Lys) variant were minimally verbal (see the speech and language section; the relevant data on this from the other two individuals were unavailable). All seven individuals showed motor delay, but the degree was much milder compared to the aforementioned cases. All those aged four years and older were able to walk without support, although one individual walked only limited distances. One individual of 3.5 years did not walk yet. Absence seizures were noted in two of these cases. One individual had asplenia. One individual had a non-progressive heart tumour of unknown origin. These individuals did not show visually recognisable or similar facial features (Fig. 1), nor did they show overlapping facial features with either classical SGS or SETBP1-haploinsufficiency disorder based on observations of clinicians. The individuals with inherited variants located furthest away from the degron and the SKI domain [c.1332 C > G p.(Ser444Arg) and c.1970T > C p.(Val657Ala)] showed mild intellectual disability or a low non-verbal IQ. In both cases, parents carrying the variant were similarly affected. For the remainder of the

study, we therefore did not distinguish these inherited variants from the de novo variants outside the degron. All variants outside the degron were considered in functional assays as one group vs those within the degron (causing classical SGS).

Variants located outside the SKI domain (n = 4) were associated with a more variable clinical phenotype. One individual with an in-frame deletion removing a threonine residue c.2885_2887del(CCA) p.(Thr962del) showed a severe phenotype with severe speech delay, inability to walk and tonic-clonic seizures. This individual had bilateral ptosis and had surgery on the right upper eyelid and left strabismus surgery. The proband's facial features appeared similar to those of the individual with the c.2984 C > T p.(Leu957Pro) variant (Fig. 1b). They both showed a round face, blepharophimosis, hypertelorism and a short nose with a bulbous tip, features also often noted in individuals with SETBP1-haploinsufficiency disorder. The latter individual had a less severe neurodevelopmental phenotype. This individual started to walk at 22 months and was able to use sign language at the age of three years.

Next, we sought to use a quantitative method to better understand the phenotypic differences of individuals with SETBP1 variants and to aid diagnosis. We therefore utilised PhenoScore[31] to quantitatively investigate whether individuals included in this study were more similar to individuals with SGS or to individuals with SETBP1-haploinsufficiency disorder. PhenoScore is an artificial intelligence-based phenomics framework that combines state-of-the-art facial recognition technology with analysis of phenotypic data in Human Phenotype Ontology (HPO) terms to quantify phenotypic similarity. We first performed a subgroup analysis to demonstrate that PhenoScore was able to distinguish SGS and SETBP1-haploinsufficiency disorder (phenotypic data and facial photographs of five individuals from each group) (Fig. 1c). We then generated individual predictions for all individuals with a missense variant in SETBP1 outside the degron included in this study using this trained PhenoScore model, to determine whether these were more phenotypically similar to the individuals with SETBP1-haploinsufficiency disorder (score = 0) or with SGS (score = 1). We performed this prediction for facial features and HPO terms separately, and also as a combined prediction (PhenoScore). Intriguingly, when only considering HPO terms alone, the majority of the individuals with variants outside the degron were more similar to those with haploinsufficiency disorder, or did not match with either SGS or haploinsufficiency disorder. However, the facial features of those with a variant directly adjacent to the degron were more similar to SGS, while those further away from the degron were more similar to haploinsufficiency disorder, suggesting a gradient for craniofacial feature formation. Interestingly, none of the tested variants outside the degron showed a PhenoScore similar to SGS; instead, the majority were either classified as no match, while a few were classified as

haploinsufficiency disorder (Fig. 1d, Supplementary Fig 1b, c, and Supplementary Data 2). Thus, the clinical presentations of variants outside the degron were distinct from classical SGS and sometimes even from truncating variants, as could be distinguished using quantitative phenotypic features.

### Individuals carrying *SETBP1* variants outside the canonical degron show generally low speech/language ability

Speech/language data were available from seven individuals in this cohort, four of whom carry a variant within the SKI domain close to the degron (probands 3, 6, 8 and 14), while three individuals harbour a variant located far from the degron and outside the SKI domain (probands 1 and 18, and the affected mother of proband 1). In this cohort, speech development during infancy was characterised by limited babbling. A history of early feeding difficulties was also present for two children (probands 3 and 18). Language ability was generally low across the group ($n = 7$) for all subdomains, including expressive, receptive, written, and social language (Table 1 and Fig. 1e). The youngest children (<9 years of age; probands 3, 6, 8 and 18) present with a severe speech and language impairment. They are minimally verbal, defined as the presence of less than 50 spoken words (Table 1), and augment verbal communication with sign language, gestural communication and digital devices. The speech motor system is impaired across all individuals in the group, with CAS the most common diagnosis ($n = 5$), followed by mild dysarthria (proband 1 and affected mother) (Table 1). CAS features included hesitancy, groping, inconsistency of production, increased errors with increasing word length, simplified word and syllable structures relative to age, and vowel and prosodic errors. Dysarthria was typically characterised by a slower rate of speech, imprecision of consonants, altered nasal resonance and monotonous speech. The adult participants (proband 1 and affected mother) had a history of poor speech development but are now able to hold appropriate conversations and speak in full sentences with speech that is usually to always intelligible. All individuals in this cohort who performed speech/language assessment are attending (probands 3, 6, 8, and 18), or had attended (probands 1 and 14, and the affected mother of proband 1), speech therapy.

### *SETBP1* variants outside the canonical degron are predicted to be damaging and cluster in the SKI domain

We used an array of computational tools to predict the functional effects of the observed *SETBP1* variants. Among the 14 variants observed in 18 individuals, eight were located in the SKI domain while six were outside any known functional domain (Fig. 1a). Using a spatial clustering analysis[32], we showed that these variants outside the degron significantly clustered in exon 4 of the canonical *SETBP1* transcript (NM_015559.2) corresponding to the SKI domain (corrected *p*-value = 9.99e-09, Bonferroni correction). All of the mutated amino acid sites were highly conserved across species with the exception of the threonine residue at position 962, which was conserved only in mammals (Supplementary Fig. 2a). All observed variants were predicted to be (likely) pathogenic by PolyPhen-2 and/or SIFT, and showed CADD-PHRED scores above 21 (Supplementary Data 3).

We went on to use the MetaDome web tool (v.1.0.1) (Supplementary Fig. 2b, c) and MTR viewer (Supplementary Fig. 2d) to visualise all *SETBP1* missense variants in the tolerance landscape of the gene. Variants in the SKI domain are located in the regions of high intolerance, while the remaining variants, including those adjacent to the HCF1-binding site, are located in the less tolerant regions (Supplementary Fig. 2b–d). Of note, *SETBP1* does not have a particularly high *Z*-score (1.1) for missense variants in the gnomAD database (v.2.1.1), indicating that the complete coding region of this gene is not extremely intolerant to missense variation. This observation is consistent

**Table 1 | Summary of speech phenotypes in individuals with *SETBP1* missense variants outside the degron**

| Nr | HGVSc (NM_015559.2) | Protein Effect (NP_056374.2) | Speech diagnosis | Minimally verbal (<50 spoken words) | Intelligibility (individual is understood) | Language[a] | | | |
| | | | | | | Expressive | Receptive | Written | Social |
|---|---|---|---|---|---|---|---|---|---|
| 1 | c.1332 C > G | p.(Ser444Arg) | Mild Dysarthria | – | Always | Low | Low | Low | Mod. low |
| Affected mother of 1 | c.1332 C > G | p.(Ser444Arg) | Mild Dysarthria | – | Usually-always | Mod. low | Mod. low | Mod. low | Average |
| 3 | c.2561 C > T | p.(Ser854Phe) | Severe CAS | + | Rarely-sometimes | Low | Low | Low | Low |
| 6 | c.2572 G > A | p.(Glu858Lys) | Severe CAS | + | Rarely-sometimes | Low | Low | Mod. Low | Low |
| 8 | c.2572 G > A | p.(Glu858Lys) | Severe CAS | + | Sometimes | Low | Low | Low | Low |
| 14 | c.2621 A > G | p.(Asp874Gly) | Severe CAS | – | Never | Low | Low | Low | Low |
| 18 | c.3499 C > A | p.(His1167Asn) | Severe CAS | + | Rarely-sometimes | Low | Low | Low | Low |

CAS Childhood apraxia of speech.
+: feature present; –: feature absent.
[a]Data from Vineland Adaptive Behaviour Scales-3 (average range 86–114); low score (≤70), moderately low score (71–85).

with the results from the MetaDome and MTR score analyses, which show that only a few regions of SETBP1 have a high intolerance for missense variants, including the part of the SKI domain in which the majority of the observed variants are located (Supplementary Fig. 2b–d). We observed that 33% of the amino acid residues that were mutated by a germline *SETBP1* missense variant were also mutated in somatic cells, particularly in haematopoietic and lymphoid cells, according to the COSMIC database (Catalogue of Somatic Mutations in Cancer v.94), including the recurrent missense variant p.(Glu858Lys). While missense variants within the degron (affecting amino acids 868-871) showed different frequencies in germline and somatic cells, consistent with the previously reported higher functional threshold in somatic cells[6], all observed missense variants outside the degron showed similar frequencies in germline and somatic cells (Supplementary Fig. 3 and Supplementary Data 4).

### *SETBP1* missense variants outside the degron cause accumulation of SETBP1 protein and increased proliferation in patient fibroblasts, independent of the SET/PP2A axis

We went on to study the functional consequences of a representative selection of variants across *SETBP1* on protein abundance and localisation, the SET/PP2A axis and cell proliferation, protein stability and degradation, and transcriptional regulation, using patient fibroblasts, as well as in HEK293T/17 cells transiently transfected with *SETBP1* expression constructs. Based on the location and their distance to the canonical degron (Fig. 1a), we included in our assays two missense variants located furthest from the SKI domain [p.(Ser444Arg) and p.(Val657Ala)], three missense variants close to the degron [p.(Glu858Lys), p.(Glu862Lys)], and p.(Leu957Pro)], and one de novo in-frame deletion [p.(Thr962del)]. In addition, we included in our assays four classical SGS variants located within the canonical degron [p.(Asp868Asn), p.(Ser869Asn), p.(Gly870Ser), and p.(Ile871Thr)] for functional comparisons to those outside the degron. In transcriptomics experiments with patient fibroblasts carrying different classes of germline *SETBP1* variants, cell lines carrying variants within the degron (classical SGS), and truncating variants (*SETBP1*-haploinsufficiency disorder) were included, again enabling functional comparisons to those outside the degron.

We first assessed the abundance of endogenous SETBP1 protein in fibroblasts derived from three patients carrying a *SETBP1* variant outside the degron. Similar to those from SGS patients, all variants outside the degron showed higher SETBP1 protein levels than healthy controls (Fig. 2a, b) while SETBP1 transcript levels did not differ (Supplementary Fig. 4a). We next assessed the abundance of FLAG-tagged SETBP1 with an expanded array of variants in transfected HEK293T/17 cells, comparing cells with variant expression constructs to those with a wild type construct. Consistent with results from patient fibroblasts, all variants showed higher SETBP1 protein levels than the wild type (Fig. 2c, d) but also higher mRNA levels (Supplementary Fig. 4b).

Increased cell proliferation has been reported in EBV-transformed lymphoblastoid cell lines (LCLs) derived from patients carrying germline classical SGS variants[6] and in leukaemic cells with somatic *SETBP1* variants that drive development of myeloid malignancies[6,33]. We therefore investigated the proliferation of fibroblasts derived from three individuals carrying a germline *SETBP1* variant outside the degron [p.(Glu858Lys), p.(Leu957Pro), and p.(Thr962del)] compared to those from sex-matched healthy controls. For two of the three variants, we observed that fibroblasts displayed significantly faster proliferation and shorter doubling time than healthy controls in a time course experiment (Supplementary Fig. 4c–f). Interaction between SETBP1 and SET has been shown to stabilise SET, protecting SET from cleavage by protease, subsequently inhibiting PP2A activity and therefore promoting proliferation in HEK and leukaemia cells[7,20,33]. We therefore examined the levels of SET in the fibroblasts. Variants outside the degron led to significantly altered SET levels in a group

comparison to controls, with differential effects within groups (Fig. 2a, e and Supplementary Fig. 4g). To determine whether SETBP1 interaction with SET was affected by the patient variants, we performed co-immunoprecipitation assays in HEK293T/17 cells co-transfected with GFP-SET and FLAG-SETBP1 variants. Although we observed more abundant GFP-SET expression with increasing SETBP1 levels when co-expressed with FLAG-SETBP1 variants [p.(Leu957Pro) and p.(Thr962del)] compared to wild type, mutated versions of FLAG-SETBP1, including those that led to faster fibroblast proliferation, retained interaction with GFP-SET similar to wild type (Supplementary Fig. 4g, h).

Unexpectedly, we saw marginal differences from wild type for cell proliferation (Supplementary Fig. 4d-f1A and B) and interaction with SET (Supplementary Fig. 4h, i) in cells carrying a recurrent missense variant [p.(Glu858Lys)] which has also been reported in leukaemia cells in atypical chronic myeloid leukaemia (aCML) patients[7]. Moreover, we did not find any differences in PP2A/PP2A phosphorylation between patient and control fibroblasts (Fig. 2a, f), further suggesting that the aetiology involving identical variants in germline and somatic cells is likely to be cell-type specific. Overall, a subset of germline *SETBP1* variants outside the degron leads to the accumulation of SETBP1 protein and promotes cell proliferation via a mechanism that is not driven by alterations in SETBP1/SET/PP2A interaction.

### *SETBP1* missense variants outside the degron cause variable disruption of protein degradation, in contrast to accumulating stable SGS variants

Next, we hypothesised that missense variants outside the degron might alter SETBP1 protein stability independent of mRNA levels. We therefore treated control and patient fibroblasts carrying variants outside the degron with cycloheximide to inhibit translation and examined the SETBP1 protein level. When treated with cycloheximide, two out of three variants [p.(Glu858Lys) and p.(Leu957Pro)] showed reduced degradation when treated with cycloheximide (Fig. 3a), similar to what was previously reported for SGS variants[6], while p.(Thr962del) showed normal degradation following cycloheximide treatment, similar to controls (Fig. 3a). To assess the impacts on a broader range of variants, we used HEK239T/17 cells transiently expressing YFP-tagged SETBP1 and treated them with cycloheximide to inhibit translation and measured relative fluorescence intensity over 24 h. We found that all classical SGS variants (within the degron) showed increased protein stability, whereas all variants outside the degron displayed similar stability to wild type (Fig. 3b and Supplementary Fig. 5). To evaluate the impact of variants on proteasome-mediated degradation, we treated HEK293T/17 cells expressing YFP-tagged SETBP1 with proteasome inhibitor MG132. Surprisingly, the classical SGS variants did not show impaired proteasome degradation, except for p.(Gly870Ser) (Fig. 3b and Supplementary Fig. 6), unlike previously reported[6]. Moreover, three variants outside the degron [p.(Ser444Arg), p.(Glu858Lys), and p.(Leu957Pro)] demonstrated disrupted proteasome degradation to various extents (Fig. 3b and Supplementary Fig. 6). To assess whether the degradation of *SETBP1* variants might be compensated by other protein degradation pathways, such as mTOR-dependent autophagy, we used the autophagy inhibitor BafilomycinA1 to treat HEK293T/17 cells expressing YFP-tagged SETBP1. While the majority of variants outside the degron [p.(Ser444Arg), p.(Val657Ala), p.(Glu858Lys), and p.Leu957Pro)] differed significantly in degradation via autophagy, most of the SGS variants were similar to wild type (Fig. 3b and Supplementary Fig. 7). These results further suggested a mechanism only partially overlapping with classical SGS. It is noteworthy that the direction and extent of degradation of the variant proteins were variable and appeared to depend on the distance of the variant from the degron region. Interestingly, protein stability and degradation of the in-frame deletion p.(Thr962del) were not affected (Fig. 3a, b), suggesting that

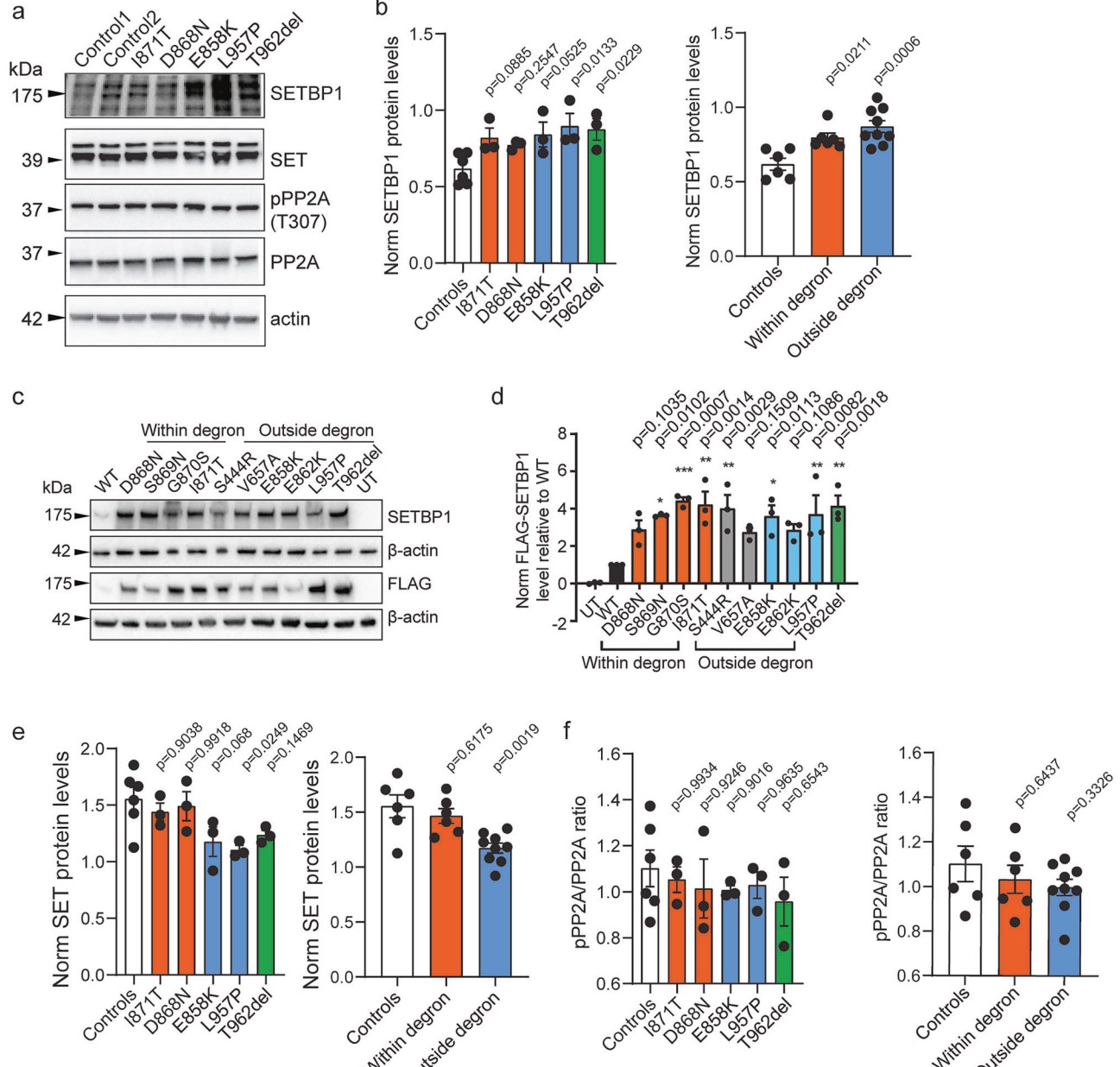

**Fig. 2 | *SETBP1* variants outside the degron show increased protein abundance but do not affect SET binding or pPP2A/PP2A ratio in patient fibroblasts.** **a** Immunoblot of whole cell lysates of control and patient human dermal fibroblasts (HDF) probed with anti-SETBP1, anti-SET, anti-pPP2A (T307) and anti-PP2A antibodies. β-actin was used as a loading control. **b** Quantification of protein levels of SETBP1 variants normalised to β-actin (right). Bars represent the mean ± SEM of three independent experiments (vs controls, one-way ANOVA and a *post-hoc* Dunnett's test). **c** Immunoblot of whole cell lysates of HEK293T/17 cells expressing FLAG-tagged SETBP1 variants probed with anti-SETBP1 and anti-FLAG antibodies. β-actin was used as a loading control (left). Representative blots of three independent experiments are shown. **d** Quantification of protein levels of FLAG-tagged SETBP1 variants normalised to β-actin (right). Values are expressed relative to wild type (WT) and represent the mean ± SEM of three independent experiments (\**p* < 0.05, \*\**p* < 0.01, \*\*\**p* < 0.001, using one-way ANOVA and a *post-hoc* Dunnett's test). **e** Normalised *SET* transcript expression (bottom) in control and patient HDFs. Bars represent the mean ± SEM of three independent experiments (vs controls, one-way ANOVA and a *post-hoc* Dunnett's test). **f** Quantification of pPP2A/PP2A ratio. Bars represent the mean ± SEM of three independent experiments (vs controls, one-way ANOVA and a *post-hoc* Dunnett's test). Source data are provided as a Source Data file. All details of statistical tests and *p*-values are provided in Source Data 4.

this variant might operate via a different pathophysiological mechanism, in spite of increased abundance.

In silico modelling of germline variants within the canonical degron that cause classical SGS has suggested effects on the interaction between the degron of β-catenin, which has a similar sequence to the βTrCP1 binding site in the SETBP1 degron, and ubiquitin E3 ligase βTrCP1[6]. To investigate whether the observed differences in protein degradation were due to alterations in SETBP1 ubiquitination, we performed immunoprecipitation of FLAG-SETBP1 and assessed its

ubiquitin level. Even though impaired proteasome degradation and autophagy were observed in two classical SGS variants and several variants outside the degron (Fig. 3b), ubiquitination was not significantly reduced in the majority of the tested variants [p.(Gly870Ser), p.(Glu858Lys), p.(Leu957Pro), and p.(Thr962del)] (Fig. 3c, d). Intriguingly, variants furthest from the degron showed significantly lower SETBP1 ubiquitination compared to wild type (Fig. 3c, d), consistent with the degradation assay results (Fig. 3b). Although based on in silico modelling of interaction with ubiquitin E3 ligase βTrCP1[6], the classical

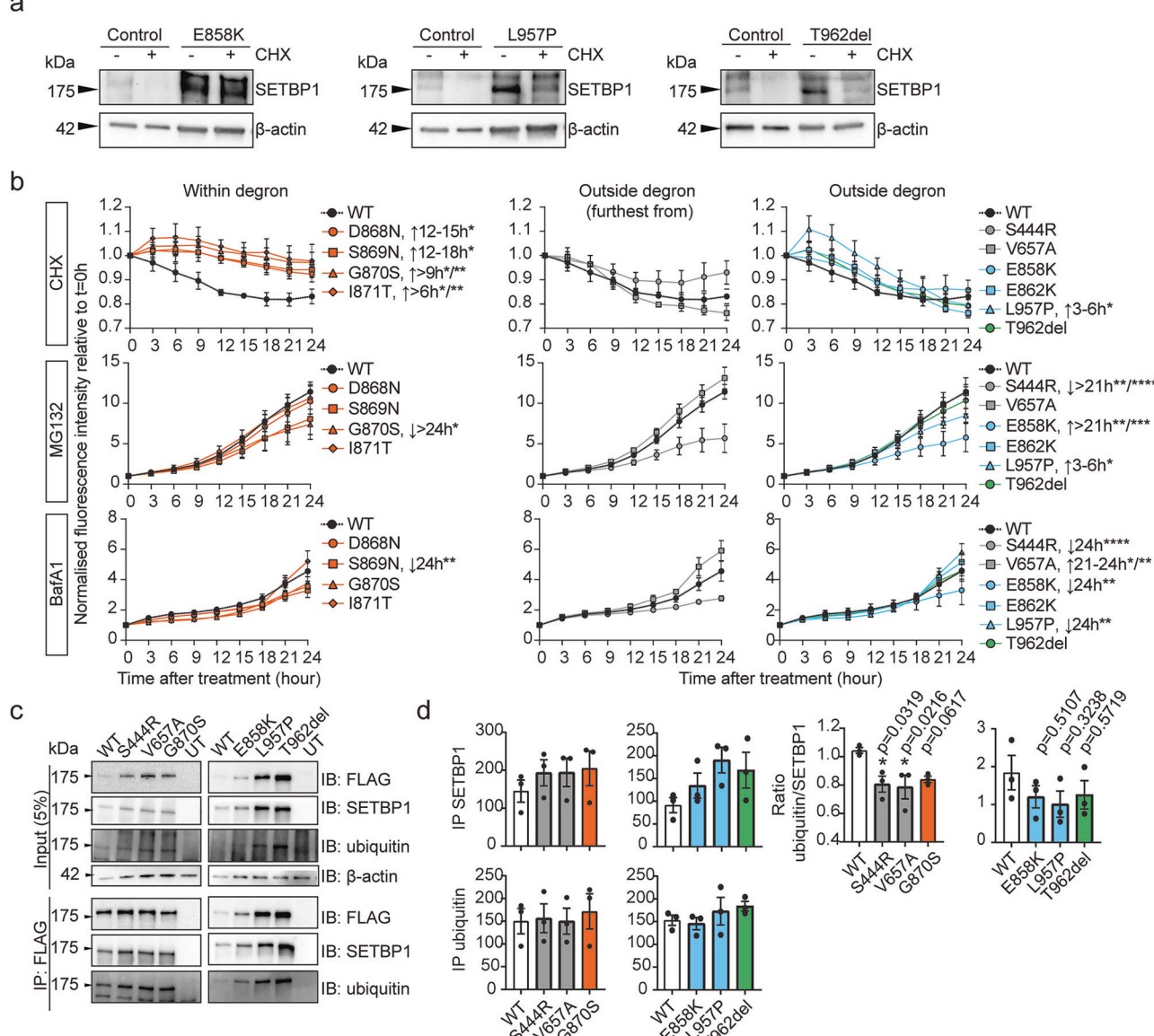

**Fig. 3 | Variable impairment of SETBP1 degradation via proteasome and autophagy pathways is associated with partial reduction of ubiquitination.**
**a** Fibroblasts derived from healthy controls and patients carrying *SETBP1* variants outside the degron were treated with a translation inhibitor, cycloheximide (CHX; 100 μg/ml) or vehicle control DMSO for 4 h. Immunoblots of whole cell lysates probed with an anti-SETBP1 antibody were shown. β-actin was used as a loading control. Results are representative of three independent experiments. **b** Relative fluorescence intensity of YFP-tagged SETBP1 variants overexpressed in HEK293T/17 cells treated with translation inhibitor cycloheximide (CHX; 50 μg/ml; top), proteasomal degradation inhibitor MG132 (5 μg/ml; middle), or autophagy inhibitor Bafilomycin A1 (BafA1; 100 nM; bottom). An equal volume of DMSO was used as a vehicle control. Fluorescence intensity was measured for 24 h with 3-h intervals and normalised to an mCherry transfection control. Values are expressed relative to

$t = 0$ h and represent the mean ± SEM of three independent experiments, each performed in triplicate ($^*p < 0.05$, $^{**}p < 0.01$, $^{***}p < 0.001$, $^{****}p < 0.0001$; two-way ANOVA and a *post-hoc* Dunnett's test). Details of statistical tests and *p*-values are provided in Source Data 4. **c** Immunoprecipitation of FLAG-tagged SETBP1 wild type and variants using FLAG-conjugated magnetic agarose and blotted with an anti-FLAG, anti-SETBP1 or anti-ubiquitin antibody. β-actin was used as a loading control. **d** Quantification of SETBP1 (top), ubiquitin (middle) in FLAG-IP fractions for inherited (outside degron, grey) and SGS variants (orange, left), and de novo variants outside the degron (right). Ratio of ubiquitin/SETBP1 in the FLAG-IP fractions was plotted (bottom). Bars represent the mean ± SEM of three independent experiments ($^*p < 0.05$, vs WT; one-way ANOVA and a *post-hoc* Dunnett's test). Details of statistical tests and *p*-values are provided in Source Data 4. Source data are provided as a Source Data file.

SGS variant p.(Asp868Asn) would be expected to show the strongest disruption in SETBP1 degradation, followed by p.(Gly870Ser), we did not see such a pattern in our proteasome and autophagy inhibition assays, nor in the level of ubiquitination. Taken together, these results suggest that accumulations of SETBP1 protein observed for a subset of variants are caused by variable disruptions in SETBP1 protein degradation via the proteasome and autophagy pathways. Other mechanisms are likely to contribute to higher SETBP1 protein levels in addition to pathways involving ubiquitination.

## *SETBP1* variants outside the degron, but not SGS variants, reduce binding to AT-rich DNA sequences and transcriptional activation

SETBP1 can bind to genomic DNA via its AT-hooks and function as a regulator of transcription[22]. We went on to assess the effects of SETBP1 variants on the capacity of the protein to bind to AT-rich DNA sequences, using a luciferase reporter system. We generated two luciferase reporters, respectively carrying six repeats of the previously reported consensus AT-rich DNA binding sequences of SETBP1 (5′-

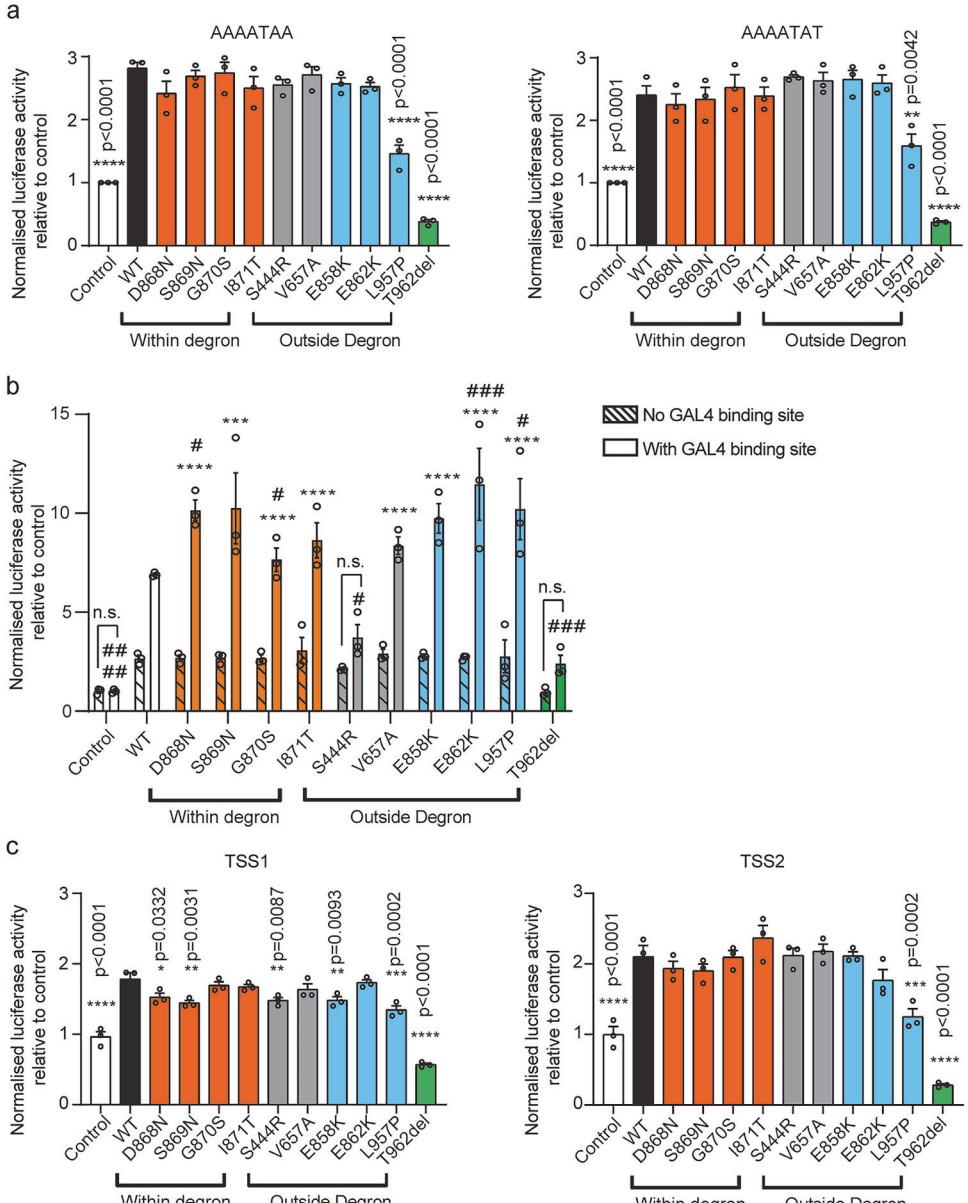

**Fig. 4 | Genotype-specific reduction of binding capacity to AT-rich DNA sequences and transcriptional activation for *SETBP1* variants outside the degron. a** Results of luciferase assays with constructs containing WT and *SETBP1* variants, and the reporter constructs with the consensus SETBP1 binding sequences. Values are expressed relative to the control condition, which used a pCMV-YFP construct without SETBP1. **b** Results of the M1H assay for SETBP1 transcriptional regulatory activity with WT and *SETBP1* variants fused with an N-terminal GAL4 in combination with a reporter construct with or without the GAL4-binding site. Values are expressed relative to the control condition, which used a pBIND2-GAL4 construct without SETBP1. **c**) Results of luciferase assays with constructs containing WT and *SETBP1* variants, and reporter constructs with *FOXP2* promoters: TSS1 (left)

and TSS2 (right). Values are expressed relative to the control condition, which used a pCMV-YFP construct without SETBP1. All graphs for luciferase assays show the mean ± SEM of three independent experiments, each performed in triplicate ($^{*}p < 0.05$, $^{**}p < 0.01$, $^{***}p < 0.001$, $^{****}p < 0.0001$ vs WT; one-way ANOVA and a *post-hoc* Dunnett's test). All graphs for the mammalian-one-hybrid assay show the mean ± SEM of three independent experiments, each performed in triplicate ($^{*}p < 0.05$, $^{***}p < 0.001$, $^{****}p < 0.0001$ vs reporter without GAL4-binding site; $^{\#}p < 0.05$, $^{\#\#\#}p < 0.001$, $^{\#\#\#\#}p < 0.0001$ vs WT; two-way ANOVA and a *post-hoc* Tukey's test). Details of statistical tests and *p*-values are provided in Source Data 5. Source data are provided as a Source Data file.

AAAATAA-3′ or 5′-AAAATAT-3′)[22]. The majority of the variants tested could still bind to AT-rich DNA sequences (Fig. 4a and Supplementary Fig. 8a, b). Of note, both variants [p.(Leu957Pro) and p.(Thr962del)] located close to the HCF binding domain (amino acids 991–994) showed significantly reduced AT-rich sequence binding capacity (Fig. 4a).

We next used a mammalian one-hybrid (M1H) assay to further delineate whether SETBP1 can induce transcriptional activity in the proximity of promoter regions without direct DNA binding. Wild-type SETBP1 fused with GAL4 showed significantly increased luciferase activity compared to empty controls and a reporter construct without

a GAL4-binding site (Fig. 4b). These results confirmed the capacity of the protein to activate transcription in the vicinity of a promoter region without direct binding to DNA, consistent with its role as a chromatin remodeller. The majority of the variants could activate transcription (Fig. 4b). Interestingly, two SGS variants and two variants close to the degron [p.(Glu862Lys) and p.(Leu957Pro)] showed significantly higher transcriptional activity compared to wild type (Fig. 4b). In contrast, the two variants furthest from the degron [p.(Ser444Arg) and p.(Thr962del)] failed to activate transcription, appearing to be LoF (Fig. 4b).

A previously published chromatin immunoprecipitation sequencing (ChIP-Seq) dataset showed that *FOXP2*, rare genetic disruptions of which lead to CAS[34–36], was one of the 70 putative SETBP1 targets in HEK cells expressing a wild-type SETBP1 construct[22]. We therefore first validated *FOXP2* as a novel direct transcriptional target of SETBP1 and demonstrated that it could be activated by SETBP1 at two different sites using a luciferase reporter assay (Fig. 4c). Moreover, most of the variants that we tested led to reduced *FOXP2* transcription activation (Fig. 4c). Notably, p.(Thr962del), which lacks only one threonine residue in the encoded SETBP1 protein, resulted in complete loss of function in all of our luciferase reporter assays, highlighting the importance of this residue for SETBP1 transcriptional activity. SETBP1 has been shown to be a largely nuclear protein, and so its potential mislocalization could lead to disruption of its function. However, we found that all *SETBP1* variants localized to the nucleus as puncta similar to wild type when assessed in transiently transfected HEK293T/17 cells (Supplementary Fig. 9a, b) and endogenously in patient fibroblasts (Supplementary Fig. 9c). Taken together, these data suggest that pathogenic *SETBP1* variants outside the degron reduce AT-rich DNA binding capacity and transcriptional activity of SETBP1 within the nucleus while preserving gross intracellular localization.

### Patient fibroblasts carrying germline *SETBP1* variants outside the degron show transcriptomic profiles that partially overlap with those from classical SGS and *SETBP1*-haploinsufficiency disorder patients

To assess whether the observed *SETBP1* variants lead to a distinct gene expression signature compared to wild type and patients with SGS and haploinsufficiency disorder, we performed RNA-seq on fibroblasts derived from two individuals carrying germline *SETBP1* variants within the degron/SGS, six with variants outside the degron, and three with truncating variants (Fig. 5a). Cells from eight healthy donors were included as controls (Fig. 5a and Supplementary Data 5). Principal component analysis (PCA) revealed that transcriptomic profiles of patient fibroblasts formed separate clusters with those from healthy individuals (Fig. 5b and Supplementary Fig. 10a). Fibroblasts carrying different types of *SETBP1* variants formed a continuum, with those outside the degron sitting between within-degron/SGS and truncating variants (Fig. 5a). Thus, variants outside the degron showed partial transcriptomic overlaps with the two clinically distinct conditions from the prior literature. We then performed differential gene expression analysis on these three conditions, comparing each to healthy controls. This analysis identified 403 differentially expressed genes (DEGs) in fibroblasts with variants within the degron, 206 DEGs in those with variants outside the degron, and 452 DEGs in those with truncating variants, after filtering for genes with low expression ($p < 0.05$, FDR; $\log_2$ fold change $\leq -1$ or $\geq 1$) (Fig. 5c). Comparison of DEGs identified in fibroblasts carrying different types of *SETBP1* variants revealed a number of DEGs present in all two or more conditions but also those unique to each variant type (Fig. 5d), further suggesting partially overlapping mechanisms. We next performed gene ontology enrichment analyses[37,38] of the consistent DEGs using an R package topGO[39] to delineate the most relevant biological processes, molecular functions, and cellular components. Functional annotation demonstrated over-representation ($p < 0.05$, Benjamini-Hochberg FDR) of an array of ontologies, some of which were partially overlapping and others of which were unique to different variant types (Fig. 5e), further supporting different pathophysiological mechanisms among SETBP1-related disorders. Several direct transcriptional targets of SETBP1 (*MECOM, RUNX1, HOXA9, HOXA10* and *MYB*) have shown differential expression in leukaemia cells from aCML patients[22–25] and in HEK cells overexpressing the p.(Gly870Ser) variant[22]. However, we did not observe differential expression of these genes in our RNA-seq data, further suggesting that the aetiological pathways are likely to be cell-type specific. To identify overlap between gene-expression signatures

of the different variant groups, we used rank–rank hypergeometric overlap (RRHO) analysis[40,41]. RRHO is a threshold-free algorithm that measures the degree of overlap by stepping through two gene lists ranked by the degree of differential expression. These analyses confirmed that DEGs of within-degron variants showed only weak overlap with those of truncating variants. DEGs of outside-degron variants showed moderate overlap with those of truncating variants and only mild overlap with those of within-degron/SGS variants (Fig. 5f and Supplementary Fig. 10b). Taken together, *SETBP1* variants outside the degron are associated with transcriptomic profiles that partially overlap with those of classical SGS variants within the degron, and truncating variants. Different variant types are linked to differences in gene ontologies, suggesting that there are not only shared aetiological mechanisms across different SETBP1-related disorders, but also mechanisms that are distinct.

### *SETBP1* variants cause differential effects on differentiation capacity, morphologies, and transcriptomic profiles in induced neurons

To investigate whether and how *SETBP1* variants affect differentiation and properties of human neurons, we differentiated patient fibroblasts carrying different types of variants into neurons using small molecules[42] (Fig. 6a, b). We selected two variants from each group and assessed differentiation capacity. Compared to control fibroblasts, different groups of variants showed different neuronal differentiation capacity (Fig. 6c, e). While within-degron/SGS and outside-degron variants showed variable differentiation capacity, truncating variants consistently showed lower differentiation capacity into Tuj1-positive neurons (Fig. 6c, d). Although the differentiation capacity within groups was variable, neuronal morphologies within groups were more consistent. Induced neurons carrying within-degron variants had more mature morphologies, i.e. bipolar neurons with larger soma and longer neurites (Fig. 6e–g and Supplementary Fig. 11). Neurons with outside-degron variants showed normal soma but longer neurites with more branches (Fig. 6e–g), while those carrying truncating variants had normal soma and a multipolar morphology (Fig. 6e–g). Of note, Sholl profiles of neurons with outside-degron variants were significantly different from those with within-degron variants ($p < 0.001$) but not with truncating variants ($p = 0.996$), while neurons with within-degron variants were significantly different from truncating variants ($p < 0.001$) (all with one-way ANOVA and a *post-hoc* Tukey's test) (Fig. 6g and Supplementary Fig. 11).

We next performed RNA-seq to assess the global transcriptomic profiles of these cells. Day 0 fibroblasts formed separate clusters from induced neurons (days 10 and 12) (Supplementary Fig. 12), consistent with successful differentiation. Since days 10 and 12 neurons are similar, we processed them as one group ("d10/12 neurons") in downstream analyses. PCA of d10/12 neurons showed that PC1 explained 45% of the variance, mainly driven by variant groups and genotypes (Fig. 7a and Supplementary Fig. 13a). The three variant groups formed largely different clusters. The two controls also formed separate clusters, which could be due to differences in sex. We then performed differential gene expression analysis on these three conditions, comparing each to healthy controls, while regressing out effects caused by sex, batch, and days in vitro. This analysis identified different numbers of significant DEGs for different variant types (within-degron variants: 969; outside-degron variants: 1050; truncating variants: 491; $p < 0.05$, FDR; $\log_2$ fold change $\leq -1$ or $\geq 1$), after filtering for genes with low expression (Fig. 7b). Comparison of significant DEGs identified in induced neurons carrying different types of *SETBP1* variants revealed a subset of DEGs present in all, two or more conditions but also some that were unique to each variant type (Fig. 7c), further suggesting partially overlapping mechanisms. RRHO analysis showed the weakest overlap between within-degron and truncating groups, as expected, since these two variant groups showed

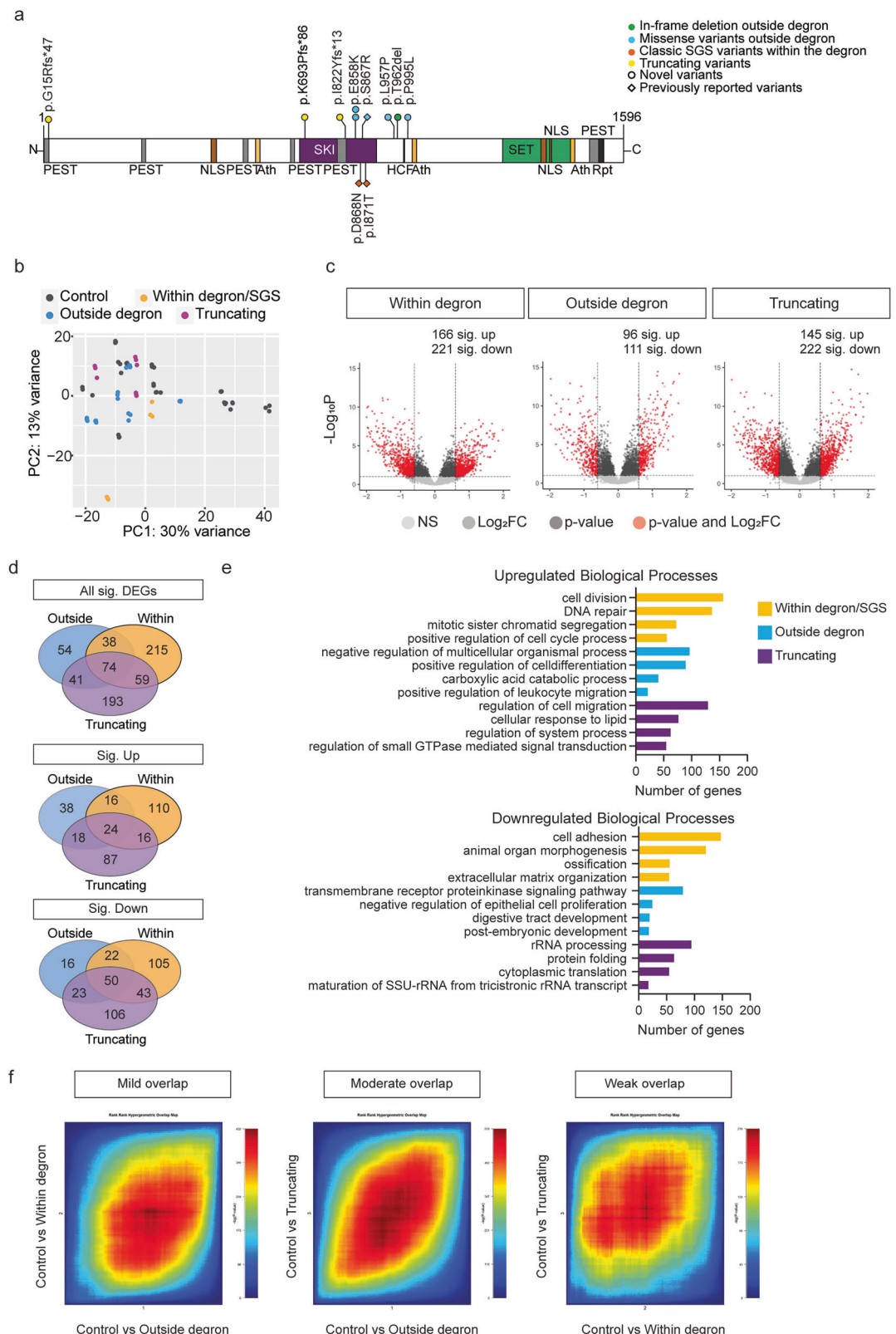

very different clinical and cellular phenotypes across measures (Supplementary Fig. 13b). We observed moderate overlap in DEGs identified in induced neurons carrying outside-degron and within-degron/SGS variants. There was overall weak overlap in DEGs identified in induced neurons with outside-degron and truncating variants, but the top up- and down-regulated genes were similar (Supplementary Fig. 13b). This overall weak overlap in DEGs could result from the

significantly reduced differentiation, i.e. the lower proportion of neurons in the truncating variant condition as reflected by the results of cell counting (Fig. 6d). Using gene ontology enrichment analyses[37,38], we uncovered partially overlapping over-represented ontologies ($p < 0.05$, Benjamini-Hochberg FDR) and also some that were uniquely over-represented in different variant types (Fig. 7d). The significant DEGs of each of the three variant types were significantly enriched for

**Fig. 5 | Fibroblasts carrying *SETBP1* variants outside the degron showed distinct transcriptomic profiles from healthy controls, SGS and *SETBP1*-haploinsufficiency disorder patients. a** Schematic representation of the SETBP1 protein (UniProt: Q9Y6X0) indicating the locations of variants included in the RNA-seq experiment. **b** Fibroblasts derived from patients harbouring a *SETBP1* variant (missense or in-frame deletion) outside the degron formed separate clusters from healthy controls, and partially overlapped with those from SGS and *SETBP1*-haploinsufficiency disorder patients. Principal component analysis (PCA) plot of variance distribution of eight control fibroblast lines (black), two lines carrying a variant within the degron (orange), six lines carrying a variant outside the degron (blue), and three lines carrying a truncating variant (purple); three technical replicates were included for all patient lines and control lines 5–8; six replicates were included for control lines 1–4. A list of cell lines included in the RNA-seq experiment can be found in Supplementary Data 5. Principal components (PC) 1 and PC2 accounted for 30% and 13% of total variance, respectively. **c** Volcano plots showing the gene expression data of control vs patient fibroblast lines (left: within degron; middle: outside degron; right: truncating) as a function of log ratios and mean average gene counts. The number of significant DEGs that are up- and down-regulated are shown ($\log_2$ fold change $\geq 1$ or $\leq -1$, $p < 0.05$, multiple testing correction with FDR). **d** Venn diagram showing the overlap of genes that demonstrated significant differential expression ($p < 0.05$, multiple testing correction with FDR) in control vs SGS variants within degron (orange), variants outside degron (blue) and truncating variants (purple). **e** Top four dysregulated GO biological processes revealed by over-representation analysis of the up- (top) and down-regulated (bottom) DEGs in patient fibroblasts carrying different *SETBP1* variants ($p < 0.05$, multiple testing correction with FDR; orange: within degron; blue: outside degron; purple: truncating). **f** RRHO heatmaps showed transcriptomic overlaps between two comparisons. Bottom left corner of the heatmap indicates overlap signal from up-regulated genes in both conditions. The top right corner of the heatmap indicates an overlap signal from down-regulated genes in both conditions. Source data are provided as a Source Data file (Source Data 6–8).

genes associated with intellectual disability (within-degron variants: $p = 3.23e-07$; outside-degron variants: $p = 3.07e-06$; truncating variants: $p = 1.12e-04$; all Fisher's exact test) but not with autism (within-degron variants: $p = 0.70$; outside-degron variants: $p = 0.94$; truncating variants: $p = 0.17$; all Fisher's exact test) in the PanelApp database (intellectual disability v3.2 and autism v.0.22) (Source Data 15). Notably, *FOXP2* was among the significant DEGs in induced neurons carrying outside-degron and within-degron/SGS variants but not for those carrying truncating variants, further suggesting that the aetiological pathways are likely to be cell-type and variant-group specific. Taken together, human induced neurons derived from patient cells carrying different *SETBP1* variants are associated with partially overlapping transcriptomic profiles with shared and unique ontologies, again consistent with the notion that both common and distinct aetiological mechanisms contribute to different SETBP1-related disorders.

## Discussion
Our study placed a strong emphasis on using cell-based functional assays and transcriptomic analyses to improve the definition and understanding of neurodevelopmental disorders caused by different classes of variants within the same gene. A thorough study that investigates and clarifies the range and heterogeneity of the SETBP1-related disorders is very much needed by the communities of clinicians, geneticists, researchers, and affected families. We demonstrate that variant-specific functional follow-up is crucial to understanding the biological underpinnings of overlapping phenotypes and heterogeneity within cohorts. Work from our group and others has previously delineated the phenotypic heterogeneity within cohorts carrying germline *SETBP1* LoF variants causing *SETBP1*-haploinsufficiency disorder, which have much milder clinical correlates than classical SGS[6,15,16]. Here, we present, to our knowledge, the largest cohort of individuals who have *SETBP1* variants (missense and in-frame deletion) outside the canonical degron region, with clinical features that do not fit with the original diagnostic criteria of classical SGS. Crucially, most of the variants we described in our cohort had never before been functionally tested and were therefore deemed of uncertain clinical significance. We used a genotype-driven approach combining deep clinical and speech phenotyping with functional follow-up of specific variants in human cell lines to investigate the phenotype-genotype associations and delineate the underlying pathophysiological mechanisms. This study is the first, thus far, to use cellular assays to systematically screen *SETBP1* variants for pathogenicity, and the results validate the promise of the approach. Our findings show that the observed phenotypic heterogeneity of the clinical and speech profiles in the patient cohort are likely to reflect distinct impacts on SETBP1 protein functions, as indexed by an array of cell-based assays. In these functional screens, we included known SGS variants as a benchmark to help delineate the differences in SETBP1

protein properties compared to selected variants throughout the locus. Since the primary purpose of the study was to define the phenotypic and functional landscape of a broad range of novel variants, comparing them to two known SETBP1-related disorders, for the broader screening part of the study we chose a cellular model that allowed us to functionally test many different variants in parallel, which is currently not achievable via much more laborious induced pluripotent stem cell (iPSC)-based models. Using an array of functional assays, we showed that variants outside the degron disrupt SETBP1 protein functions via aetiological mechanisms including impairments in ubiquitination, DNA binding capacity, transcription and cell proliferation, which are independent of SETBP1 protein levels or the SET/PP2A axis. Interestingly, we found that *SETBP1* variants led to reduced transcription of its direct target *FOXP2*, of which rare variants are a known cause of monogenic speech disorder characterised by CAS[34–36]. Finally, building on the findings from our broader screen, we showed that the differentiation capacity, morphologies, and transcriptomic profiles of induced neurons derived from patient cells differ from controls, uncovering differential effects for different variant groups, along with variability within groups.

### Variable severity of broad clinical features depends on the proximity of variants to the degron
The variable severity of the broad spectrum of clinical features observed in our cohort could largely be categorised into three groups based on the distance of the variant from the canonical degron. We found that individuals with variants in close proximity to the degron within the SKI domain (affecting amino acids 862–874, excluding the degron 868–871) in our current cohort showed a more severe phenotype with severe intellectual disability and inability to speak, and are often unable to walk. These individuals are much more severely affected than those with *SETBP1*-haploinsufficiency disorder[16] and yet do not fulfil the original diagnostic criteria of classical SGS as defined by Lehman et al.[30]. These classical SGS criteria were proposed at a time when the causative gene was not yet identified, which may have led to an ascertainment bias in the first studies after identification of the gene[2,6]. Thus far, two individuals with a *SETBP1* variant within the hotspot region, who displayed an atypical milder phenotype which was not concordant with the original diagnostic criteria of SGS, have been reported. In 2020, Sullivan et al.[43] described an individual with a p.(Ile871Ser) variant who had a moderate intellectual disability, no congenital anomalies, and showed less apparent dysmorphisms than patients with classical SGS, including mild midface retrusion, hypertelorism, short upturned nose and prominent low-set ears. Recently, Beaman et al.[44] described an individual with a p.(Ile871Met) variant who had several typical SGS features, such as hydronephrosis caused by a megaureter, febrile seizures, and associated facial features. However, the individual was in good health at 12 years of age and

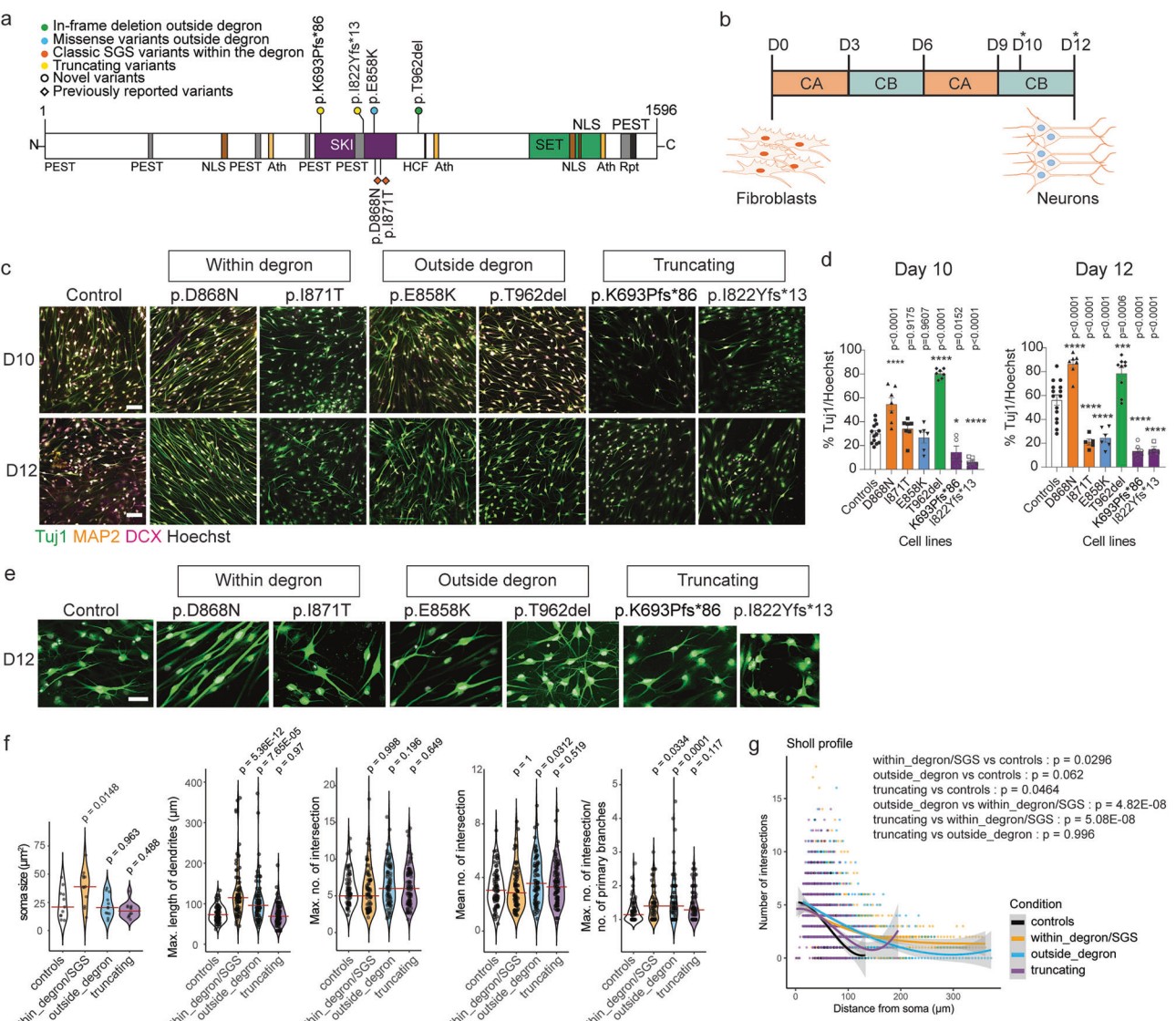

**Fig. 6 | Induced neurons carrying different *SETBP1* variant groups showed distinct differentiation capacity and morphology, and transcriptomic profiles.** **a** Schematic representation of the SETBP1 protein (UniProt: Q9Y6X0) indicating the locations of variants included. **b** Schematic representation of the generation of induced neurons from fibroblasts. Asterisks indicate the time of harvest. CA: compound set A; CB: compound set B. **c** Confocal microscopy images of immunostained neuronal markers Tuj1 (green), MAP2 (orange), and DCX (magenta) in control and patient-derived induced neurons. Nuclei were stained with Hoechst 33342 (white). Merged images are shown. Results are representative of three independent experiments. Scale bar = 100 μm. **d** Quantification of the proportion of Tuj1-positive cells (%Tuj1/Hoechst) at days 10 and 12. Bars represent the mean ± SEM of three independent experiments (vs controls; one-way ANOVA and a *post-hoc* Dunnett's test). Details of statistical tests and *p*-values are provided in Source Data 9. **e** Zoomed in confocal microscopy images of immunostained neuronal markers Tuj1 (green) in control and patient-derived induced neurons. Nuclei were stained with Hoechst 33342 (white). Results are representative of three independent experiments. Scale bar = 50 μm. **f** Quantification of neuronal morphology using Sholl analysis. Violin plots of three independent experiments (vs controls; one-way ANOVA and a *post-hoc* Dunnett's test). Brown crossbars indicate the median. At least 40 neurons per cell line from three independent differentiation experiments were measured. Details of violin plot statistics, statistical tests and *p*-values are provided in Source Data 9. **g** Sholl profile of neuronal morphology. Data are from three independent experiments. Individual data points represent the number of dendritic intersections per radial distance from the soma. Solid lines show the predicted mean per condition, fitted using cubic polynomial regression (3rd-degree) via linear modelling; shaded bands represent 95% confidence intervals around the model fit. Statistical analysis was performed using one-way ANOVA and a *post-hoc* Tukey's test. At least 40 neurons were analysed per cell line. Details of statistical tests and adjusted *p*-values are provided in Source Data 9. Source data are provided as a Source Data file.

showed less severe developmental delay compared to what would be expected in SGS. These individuals who were more mildly affected than those with classical SGS may point to a broader SGS-associated phenotype than originally defined, with a different (non-lethal) life expectancy, possibly involving interactions with other variants in the genomic background and/or depending on the specific amino acid residue changes. In addition, this wider spectrum of SGS may also be applicable to cases with aetiological variants in the 862-874 region who

displayed similar facial features, including mild midface retrusion, microcephaly and prominent low-set ears (Fig. 1).

The group of *SETBP1* variants affecting residues at positions 854-858 (located further from the degron) showed a phenotype including mild to moderate intellectual disability. Motor development was delayed, but they all achieved the milestone of walking without support, and growth was normal. Clinicians did not observe a consistent pattern of dysmorphisms (Fig. 2). The p.(Glu858Lys) variant was noted

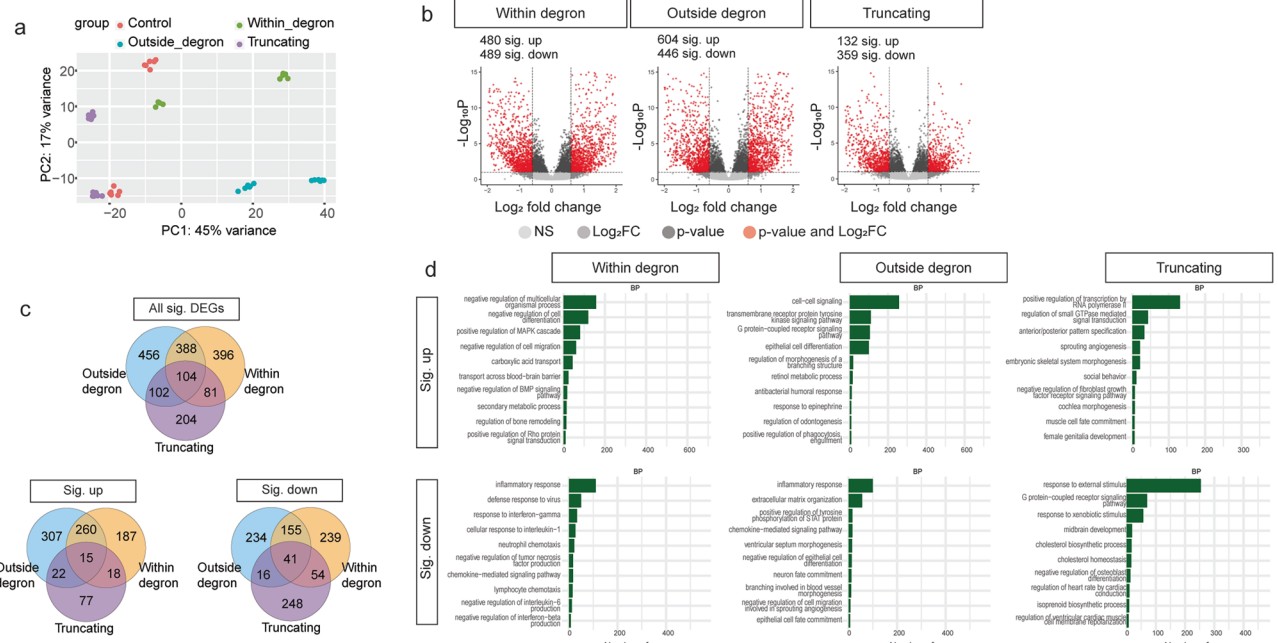

**Fig. 7 | Induced neurons carrying different *SETBP1* variant groups showed distinct transcriptomic profiles. a** PCA plot of induced neurons derived from control (red), variant within the degron (green), variant outside the degron (blue), and truncating variant (purple); two lines each from patient variant groups and control in each differentiation; three independent differentiation experiments were included for all patient lines and control lines. A list of cell lines included in differentiation and RNA-seq is included in Supplementary Data 7. **b** Volcano plots showing the gene expression data of control vs patient induced neurons (left: within degron; middle: outside degron; right: truncating) as a function of log ratios and mean average gene counts. The number of up- and down-regulated DEGs that had a log$_2$ fold change ≥1 or ≤−1 are shown ($p < 0.05$, multiple testing correction with FDR). **c** Venn diagram showing the overlap of genes that demonstrated significant differential expression ($p < 0.05$, multiple testing correction with FDR) in control vs SGS variants within degron (orange), variants outside degron (blue) and truncating variants (purple). **d** Top 10 dysregulated GO biological processes revealed by over-representation analysis of the up- (top) and down-regulated (bottom) DEGs in patient-induced neurons ($p < 0.05$, multiple testing correction with FDR; orange: within degron; blue: outside degron; purple: truncating). Source data are provided as a Source Data file. Details of statistical tests and *p*-values are provided in Source Data 10–11.

in four individuals. Leonardi et al.[45] also described an individual with the same variant, showing severe intellectual disability and epilepsy. In that study, the variant was reported to be inherited from an asymptomatic mother whose variant was de novo. Unfortunately, no other tissues were tested in the mother to exclude the possibility of mosaicism.

Individuals with variants located outside the SKI domain ($n = 4$) showed a more variable clinical phenotype, ranging from mild to severe intellectual disability. Two of these individuals [carrying p.(Leu957Pro) and p.(Thr962del) variants] showed a very similar facial phenotype, including a round face, blepharophimosis, hypertelorism and a short nose with a bulbous tip, features also often noted in individuals with *SETBP1*-haploinsufficiency disorder (Fig. 1). The two individuals with variants located furthest from the SKI domain [p.(Ser444Arg) and p.(Val657Ala)] had inherited them from affected mothers who also have low IQ (grandparents and other family members were not available for testing). We therefore also included these variants in the functional cellular assays together with the de novo variants outside the degron in our study. Indeed, these two variants had significant impacts on SETBP1 protein functions as shown in our cellular assays. In light of this, we recommend that clinicians should examine the clinical phenotypes of family members carrying the same variants, as well as perform cellular assays to test pathogenicity when assessing the pathogenicity of inherited variants of uncertain significance based on ACMG (American College of Medical Genetics and Genomics) guidelines[46]. Clinicians could also consider performing functional prediction using algorithms where available, such as those for known epilepsy ion channel genes[47]. Given the variability and broad spectrum of clinical features observed in individuals with *SETBP1*

missense variants, further studies including larger cohorts of individuals will help in further profiling of the phenotypic spectrum.

The variable phenotypic severity and partial clinical overlaps with SGS or *SETBP1*-haploinsufficiency disorder could suggest the potential existence of a third clinical entity involving disruptions of this gene. Alternatively, different types of *SETBP1* variants could lead to a continuum of clinical features of SETBP1-related disorders, with SGS and *SETBP1*-haploinsufficiency disorder positioned at two extremes of severity. With the view to aid diagnosis and enhance understanding of the phenotypic spectrum of SETBP1-related disorders, we used PhenoScore to quantify the similarity of clinical and facial features. Using a model trained with SGS and haploinsufficiency disorder, we further demonstrated that the phenotypic representation associated with variants outside the degron is variable and only partially overlapping with the two previously known disorders (Fig. 1). With this approach, we were able to distinguish the heterogeneous facial features, a task that was challenging for human eyes. Future studies, including a larger cohort whose clinical and facial features are examined in a standardised way (HPO terms and with photographs taken from different angles), will be beneficial for further delineating the phenotypic complexity of SETBP1-related disorders and profiling any dysmorphism, and thus improving deep learning-based clinical diagnosis.

In our RNA-seq experiments with patient fibroblasts, as well as induced neurons derived from patient cells that directly compare the different groups of variants (to our knowledge, the first involving this gene), the three classes of *SETBP1* variants led to distinct transcriptomic profiles affecting different biological pathways with differentially expressed genes that were only partially overlapping. This pattern of findings is further evidence in support of a possible

continuum of SETBP1-related disorders, with SGS and *SETBP1*-haploinsufficiency disorder positioned at two extremes of severity, involving pathophysiological mechanisms that are partly overlapping, along with some features that are unique to each group. Molecular investigations of future larger cohorts, including individuals with different variants in combination with genome-wide analysis, for example, DNA methylation analysis (episignatures)[48,49], will provide further clarification over the clinical definition and aetiology, as well as improving clinical diagnosis. Our participants had long-term chronic difficulties across a number of developmental domains. As early as possible following diagnosis, individuals with *SETBP1* variants outside the degron region should receive careful assessment across core domains of speech and language, attention, motor and sleep, leading to better targeted and earlier intervention to optimise children's health and developmental outcomes. Overall, based on our results, when interpreting pathogenicity of variants of unknown significance, we recommend using a wide range of prediction tools and performing functional assays. A list of prediction tools used in our study can be found in the section "Tools".

### *FOXP2* is a direct transcriptional target of SETBP1

Individuals with *SETBP1*-haploinsufficiency disorder showed significant impairments in both receptive and expressive language, suggesting SETBP1 as a strong candidate for speech and language disorders[15]. Of note, we have here demonstrated that individuals carrying *SETBP1* variants outside the degron, where speech and language data were available, showed generally low language ability for all subdomains, including expressive, receptive, written and social language. Intriguingly, our cell-based experiments identified *FOXP2* as a novel direct transcriptional target of SETBP1 (Fig. 4a). All functionally tested *SETBP1* variants outside the degron displayed significantly decreased *FOXP2* transcription in luciferase reporter assays. *FOXP2* is one of the few genes identified to date for which disruptions yield disproportionate effects on speech and language, yet little is known about its upstream regulators. Notably, *FOXP2* was a significant DEG in our patient cell-derived induced neurons but not in patient fibroblasts, or in those performed on patient leukaemia cells and cell lines expressing *SETBP1* constructs in previous studies, further pointing towards potential cell type-specific aetiological pathways. Moreover, we have recently demonstrated that the avian ortholog of *SETBP1* regulates *FoxP2* in the zebra finch, which is a well-established model of vocal learning, using a luciferase reporter assay[50]. Our results provide molecular evidence linking two strong candidate genes implicated in CAS, which might help provide insight into the severe speech and language impairments observed in our, and other, cohorts.

### A broad range of functions are disrupted by variants outside the degron via pathophysiological mechanisms independent of SETBP1 protein dosage

As seen in the summary of our functional analyses (Fig. 8), a broad spectrum of cellular functions was affected. In these cellular assays, variants outside the degron led to molecular consequences distinct from classical SGS variants. Classical SGS variants showed increased protein stability and higher SETBP1 protein levels as a result of impaired degradation by proteasome and autophagy. Moreover, while the binding capacity of SGS variants to AT-rich DNA sequences was not affected, a subset of SGS variants showed higher transcriptional activation, suggesting a gain of function. These findings are consistent with prior studies showing global upregulation of SETBP1 binding to genomic regions and increased chromatin accessibility when overexpressing an SGS variant [p.(Gly870Ser)] in HEK cells[22].

In contrast, *SETBP1* variants outside the degron demonstrated an array of functional disruptions that only partially overlapped with those observed for SGS variants. Although the majority of variants outside the degron resulted in more abundant SETBP1, there were variable extents of disruption in the degradation of this protein via the proteasome and autophagy machinery, suggesting that impaired protein degradation might not be a core pathogenic mechanism. The p.(Glu862Lys) variant that is in close proximity to the degron led to increased protein levels and transcriptional activation, similar to SGS variants, but neither protein stability nor degradation was affected. This points to a functional landscape partially overlapping with SGS variants but distinct from the other variants located further from the degron. The variant that is further away from the degron but still in the SKI domain [p.(Glu858Lys)] led to more abundant SETBP1 protein, reduced degradation via proteasome and autophagy, and lower *FOXP2* transcription. However, even though p.(Glu858Lys) showed functional results overlapping with those of SGS variants, the individuals carrying the variant still presented milder clinical features compared to classical SGS (amino acids 868-871) and to those with variants nearest to the degron (amino acids 862-867 and 873-874), as also exemplified in our PhenoScore results.

Functional impairments of variants further away from the SKI domain could be largely divided into two groups, matching the variable clinical features observed in the individuals that carried them. In our cell-based assays, two variants [p.(Leu957Pro) and p.(Thr962del)] led to a loss of DNA-binding and transcriptional activity despite the increased protein levels detected in transfected cells and patient fibroblasts, consistent with the observed clinical features being closer to individuals with *SETBP1*-haploinsufficiency disorder. These two variants are located close to the HCF binding site (amino acids 991–994), which is an important component of the COMPASS complex. Piazza et al. reported that mutant SETBP1 still retained the ability to bind to HCF, PHF6/8 and KMT2A, in HEK cells overexpressing an SGS variant [p.(Gly870Ser)][22]. This raises a hypothesis that the loss of transcriptional activity for variants such as p.(Leu957Pro) and p.(Thr962del) could result from disruption of interaction with HCF, a core protein of the SET1/KMT2A COMPASS-like complex responsible for H3K4 mono- and di-methylation, and chromatin accessibility[22,51]. In particular, the threonine residue at the 962 position is likely crucial for SETBP1 transcriptional activity, potentially via HCF1-related pathways given its proximity to the HCF-binding site and the complete loss of transcriptional activation in cells expressing p.(Thr962del). Intriguingly, variants that were furthest from the degron region [p.(Ser444Arg) and p.(Val657Ala)] showed mostly impaired SETBP1 degradation by both proteasome and autophagy due to disrupted ubiquitination. It is possible that these variants are physically closer to the degron in the three-dimensional conformation of the protein. However, we could not predict how *SETBP1* variants might affect its protein folding or interactions with interactors due to limited knowledge about the overall structure of the protein.

SETBP1 has been identified as a protein with intrinsically disordered regions (IDRs), which are regions that are prone to mutations and are often found in proteins associated with cancers[52]. Indeed, the majority of *SETBP1* missense variants identified to date, both somatic and germline, are located within an IDR (amino acids 858-880)[52]. Many proteins containing IDRs do not have a single well-defined conformation, and the conformation changes depending on interactions with cofactors. Most proteins with IDRs that are elements of the transcriptional machinery are multifunctional, involved in processes such as regulating degradation via a linear motif, binding to genomic DNA, activating/repressing transcription, and/or modulating histone modifications[52]. This wide range of functions could explain the broad spectrum of impairments found in our cell-based assays and make it more difficult to predict how variants affect SETBP1 function. Future studies that profile and delineate the structural impact of *SETBP1* variants and how they affect interactions with cofactors, for example, via proteomic approaches, will aid the understanding of their impacts on protein functions and thus aetiology.

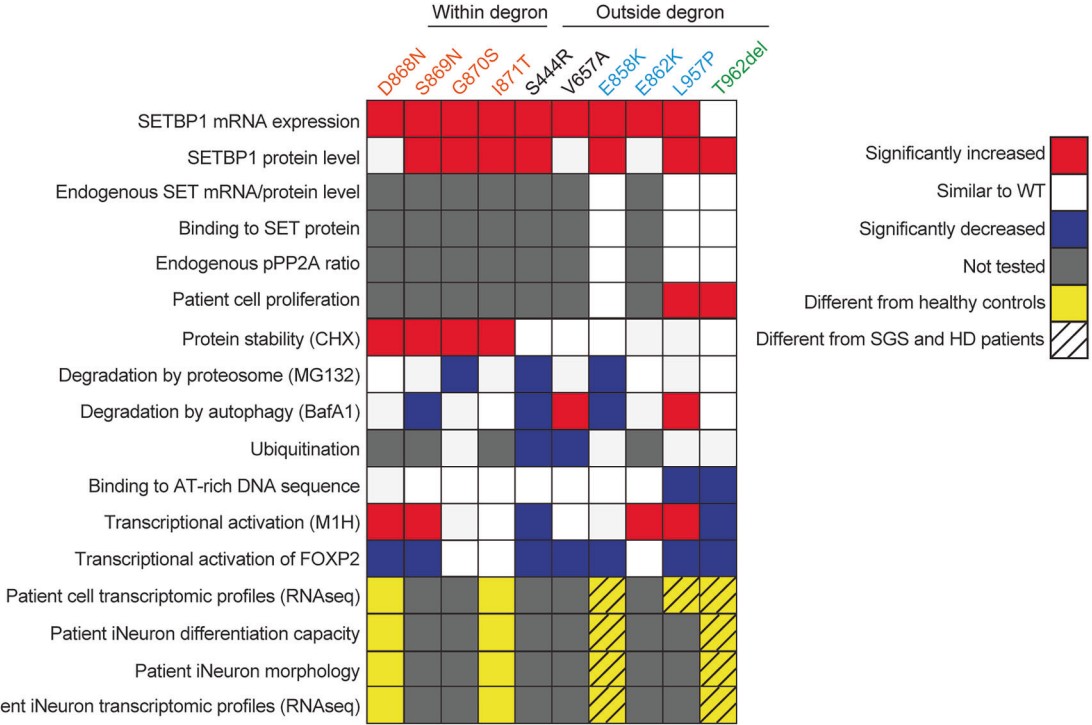

**Fig. 8 | Summary of functional assays: variants outside the degron disrupt a spectrum of SETBP1 functions by dysregulating transcription, reducing DNA binding capacity, variable protein degradation impairment and ubiquitination independent of SETBP1 level or the SET/PP2A pathway.** A heat map summarising the functional characterisation of *SETBP1* variants within (classical SGS) and outside the degron. Classical SGS variants showed increased protein stability and higher SETBP1 protein levels. While the binding capacity of SGS variants to AT-rich DNA sequences was not affected, a subset showed higher transcriptional activation, suggesting a gain of function. In contrast, *SETBP1* variants outside the degron demonstrated a broad spectrum of functional disruptions, which could be largely categorised into two groups. Although the majority of these variants resulted in more abundant SETBP1 protein, there were variable degrees of disruption in SETBP1 degradation via the proteasome and autophagy machinery. Variants furthest from the degron (grey) showed mostly reduced SETBP1 degradation by both proteasome and autophagy due to disrupted ubiquitination, while only a subset of those in the proximity of the degron demonstrated impaired degradation and mildly reduced ubiquitination. Although increased proliferation was seen for patient fibroblasts carrying two variants outside the degron, suggesting a partially overlapping mechanism with classical SGS, SET expression levels or pPP2A/PP2A ratio (indicative of PP2A activity) were not affected. Two variants outside the degron led to lower affinity to AT-rich DNA sequences, suggesting loss of function. Transcriptional activation was affected in the majority of variants outside the degron to various degrees. Patient fibroblasts carrying variants outside the degron showed different transcriptomic profiles compared to healthy controls, and to those from SGS and SETBP1-haploinsufficiency disorder patients, with differentially expressed genes involved in different gene ontology biological processes. Patient-induced neurons with different types of *SETBP1* variants showed different differentiation capacity, morphology, and transcriptomic profiles. HD: *SETBP1*-haploinsufficiency disorder.

## Cell-type-specific aetiology of *SETBP1* variants

We identified a subset of genes that are differentially expressed in at least two classes of *SETBP1* variants in our RNA-seq data, suggesting partly shared aetiological pathways with SGS or *SETBP1*-haploinsufficiency disorder, complementing the observed partial overlap in clinical manifestation. Nevertheless, we did not observe overlaps with genes dysregulated in aCML patient cells, nor did we observe any changes in proliferation in fibroblasts derived from individuals carrying p.(Glu858Lys), a recurrent variant that has also been reported in individuals with leukaemia[7]. This is consistent with the findings of a recent study where disruption of the SET/PP2A pathway was not detected in SGS neural progenitors and cortical organoids[8], suggesting that the aetiological pathways underlying germline and somatic *SETBP1* variants are likely cell type-/tissue-specific. Although patient fibroblasts are not the most severely affected cell type, they are relatively easy to assess and endogenously express SETBP1. Since our study placed a strong emphasis on using cell-based functional assays and transcriptomics to screen for pathogenicity and improve the definition of SETBP1-related disorders, we chose cellular models that allow for systematic functional screening of variants. Investigations of the spatial and temporal expression of SETBP1 and the functional impact of variants during brain development are much needed to understand the pathways that go awry. This could be achieved by using neuronal models that carry *SETBP1* variants and endogenously express SETBP1. To date, only a handful of studies have investigated the impacts of an SGS variant or CRISPR-engineered *SETBP1* deletions or truncating variants in early-stage human neural models[8,26,27]. However, the impacts of *SETBP1* missense or truncating variants have not been studied in neurons, and there has not been any direct comparison across groups of variants. Models derived from iPSCs are not suitable for functional screening of variants since the generation of cells carrying selected variants, either from patient cells or introduced using gene-editing prior to long-term culture of differentiated cells, requires a different study design, extensive quality control and more detailed investigation, which is labour-intensive. In this study, to systematically compare the neuronal impacts across variant groups, we studied a smaller selection of representative variants using a method that enables relatively rapid generation of induced neurons directly from patient fibroblasts using small molecules. Notably, we showed that the differentiation capacity, morphologies, and transcriptomic profiles of induced neurons derived from patient cells differ from controls, uncovering differential effects for different variant groups, along with variability within groups. We note that the differentiation efficiency and the neuronal subtypes obtained using this method are still relatively limited. Future work comparing the functional impacts of different types of variants in models comprising different neuronal

subtypes from different brain regions will be important to further delineate the aetiology, complexity and pleiotropy of SETBP1-related disorders. Importantly, the present study provides valuable insights into which variants should be prioritised for labour-intensive follow-up work involving a subset of variants, chosen based on the results obtained from our functional assays.

## SETBP1 is a central upstream element of biological networks disturbed in neurodevelopmental disorders

Our RNA-seq analysis of patient-induced neurons carrying different types of *SETBP1* variants revealed a number of DEGs, including transcription factors that are important for neurodevelopment and associated with intellectual disability and/or autism. Previous chromatin immunoprecipitation and mass spectrometry analyses in HEK cells overexpressing SETBP1 have shown its role as a potential epigenetic hub[22]. Moreover, SETBP1 directly regulates transcription of *FOXP2*, which has been shown to interact with an array of transcription factors implicated in neurodevelopmental disorders characterised by speech and language deficits[53,54]. Given the broad expression of SETBP1 in neural progenitors and neurons, and its roles as a chromatin remodeller and transcription factor, our findings further suggest that this protein is a central element of biological networks regulating brain development, and could be important in the pathophysiology of neurodevelopmental disorders caused by genetic disruptions of its interactors or downstream targets.

In conclusion, in this study, we investigated the genotype–phenotype associations of germline *SETBP1* variants by thoroughly profiling the clinical and speech-language phenotypes. We have established an array of functional assays for testing the pathogenicity of the identified variants, which can be applied to variants of uncertain significance in the future. Despite the partial overlap of results in clinical and functional analyses with SGS, our data suggest that *SETBP1* variants outside the degron cause a clinically and functionally variable form of neurodevelopmental disorder that is milder than SGS and, in certain aspects, more severe than *SETBP1*-haploinsufficiency disorder. In particular, we have shown impairments in ubiquitination, DNA binding capacity, transcription, and cell proliferation, depending on the proximity of variants relative to the degron in the SKI domain and the specific changes in amino-acid residues. Our quantitative approach using PhenoScore shows promise for facial feature-based clinical diagnosis, and understanding of phenotypic complexity and genotype–phenotype correlations when combined with transcriptomics of patient-derived cellular models. Overall, these findings reveal that variants outside the degron act via a range of LoF pathophysiological mechanisms independent of protein abundance, unlike SGS and *SETBP1*-haploinsufficiency disorder, providing valuable new insights into diagnosis and aetiology of SETBP1-related disorders.

## Methods

### Patients and consent

This study complies with all relevant ethical regulations. Consent procedures were in accordance with the Declaration of Helsinki and local ethical guidelines of participating centres. The Human Research Ethics Committee of The Royal Children's Hospital, Melbourne, Australia, approved study and testing on a research basis of proband 1 (Project 37353). Patient materials were obtained through international collaborations involving clinicians from various countries, and patient fibroblasts were derived according to local ethics approval (Medical Ethics Committee of the Radboud University Nijmegen Medical Centre, Nijmegen, The Netherlands). This study analysed the medical data for patients with a variant in *SETBP1*. Only individuals with a molecular diagnosis of carrying a *SETBP1* variant are included. No next-generation DNA sequencing data were generated in this study. The clinical and genotype information were obtained through international collaborations involving clinicians from various countries. Some of these collaborations were established via GeneMatcher[55]. Written and informed consents were obtained from all patients or legal guardians for participating in the study. Written and informed consent was also obtained for the publication of patient photos, where appropriate. The authors affirm that human research participants/guardians provided written informed consent, for the publication of the images in Fig. 1, and have seen the images within the context of the publication. The authors affirm that human research participants/guardians have seen the data within the context of the publication. No participant compensation was included.

### Molecular analyses

Peripheral blood samples were collected in a diagnostic context from the proband, and parents when available. Results originated from diagnostic-based or previously reported research-based genome sequencing, exome sequencing, gene panel testing for intellectual disability or epilepsy or via initial direct Sanger sequencing of the *SETBP1* gene based on clinical phenotypes, which in one case was followed by trio-based exome sequencing (details in Supplementary Data 1). Previously reported variants are indicated in Supplementary Data 1. Patients identified in a diagnostic context were not recruited for this study; consequently, no DNA sequencing was performed, and no sequencing data are available for deposition. Only clinical and genotype information were obtained. The clinical and speech phenotype data are provided in the Supplementary Data 1 and 2 and Source Data files.

### Speech and language phenotyping

The history of speech and language development was recorded. Speech was analysed for the presence of articulation, phonological disorder, stuttering, CAS[17,56], and dysarthria[57,58]. Clinical diagnostic reports from local treating speech pathologists were used to validate the diagnoses, with 100% agreement. The Intelligibility in Context Scale[59] examined how often the participant's speech was understood, with a 5-point scale of responses ranging from never to always. Children were assessed for verbal, written and social language with the standardised Vineland Adaptive Behaviour Scales-Parent/Caregiver III[60]. Adult participants were assessed for receptive language with the Peabody Picture Vocabulary Test-4[61], and clinical speech pathology (S.J.T., R.O.B., and A.T.M.) and neurology (I.E.S.) assessment determined expressive language, social skills and written language ability.

### Quantitative phenotyping using PhenoScore

To investigate whether individuals included in this study were more similar to individuals with SGS or individuals with *SETBP1*-haploinsufficiency disorder, we utilised PhenoScore[31]. PhenoScore is an artificial intelligence-based phenomics framework that combines state-of-the-art facial recognition technology with analysis of phenotypic data in HPO terms to quantify phenotypic similarity.

We first collected phenotypic data and facial photographs of five individuals with SGS and five individuals with *SETBP1*-haploinsufficiency disorder, and performed a subgroup analysis to demonstrate that PhenoScore is able to distinguish these two groups. Then, using this trained PhenoScore model, individual predictions were generated for all individuals of the study who had a missense variant in *SETBP1* outside the degron, yielding a score between 0 and 1 for each, to determine whether they are more phenotypically similar to the individuals with *SETBP1*-haploinsufficiency disorder (score = 0) or SGS (score = 1). The facial feature vectors were inspected as well, and UMAP plots were generated to visualise the facial similarity between all individuals. Code used in the PhenoScore analysis is available at: https://github.com/ldingemans/PhenoScore.

## Spatial clustering analysis of missense variants

All independently observed *SETBP1* missense variants outside the canonical degron were included in the spatial clustering analysis[32,62], which tests whether variants cluster spatially within a gene while accounting for the length of the respective gene. The geometric mean was computed over the locations of observed missense variants in the canonical transcript of *SETBP1* (9,899 bp; NM_015559.2) and subsequently compared to 100,000,000 permutations by randomly redistributing the variant locations over the total size of the *SETBP1* coding region and calculating the resulting geometric mean from each of these permutations. The corrected *p*-value was computed by checking how often the observed geometric mean distance was smaller than the permutated geometric mean distance and considered significant if <0.05. Code used in the spatial clustering analysis is available at: https://github.com/laurensvdwiel/SpatialClustering[32].

## Cell culture and transfection

HEK293T/17 cells (CRL-11268, ATCC) were cultured in Dulbecco's modified Eagle's medium (DMEM) (Gibco) supplemented with 10% foetal bovine serum (FBS) (Gibco) and 100 U/ml Penicillin-Streptomycin (Thermo-Fisher) at 37 °C and 5% $CO_2$. For immuno-fluorescence analysis, cells were seeded onto coverslips coated with 100 µg/ml poly-L-lysine (Merck-Millipore). Fibroblast cell lines were established from skin biopsies of patients with *SETBP1* variants and controls at the Cell Culture Facility, Department of Human Genetics at Radboud University Medical Centre. GM27453 fibroblast line was purchased from the NIGMS Human Genetic Cell Repository, Coriell Institute. Human dermal fibroblasts were cultured in DMEM supplemented with 20% FBS, 100 U/ml Penicillin-Streptomycin and 1% sodium pyruvate (Thermo-Fisher) at 37 °C and 5% CO2. Transfections were performed using transfection reagent GeneJuice (Merck-Millipore) following the manufacturer's instructions or poly-ethyenimine (PEI) (Sigma) in a 3:1 ratio with the total mass of DNA transfected.

## Generation of induced neurons from fibroblasts using small molecules

Human dermal fibroblasts were differentiated into induced neurons according to Yang et al.[42] with modifications. In brief, on day −1, fibroblasts were first cultured on Matrigel (1:100)-coated Ibidi 8-well removable chamber polymer slide flask with 14,684 cells in 400 µl of medium per well and 6-well plates with 15,1578 cells in 2 ml of medium per well. Cells were maintained in fibroblast medium [DMEM high-glucose medium (Invitrogen) consisting of 10% foetal bovine serum (Gibco), 1% nonessential amino acids (Gibco), and 1 mM GlutaMAX (Gibco)] for 24 h. On day 0, medium was completely replaced by neuronal induction medium (NIM) consisting of DMEM/F12:neurobasal medium (1:1) supplemented with 1% N-2, 2% B-27, 40 ng/mL brain-derived neurotrophic factor, 20 ng/mL glial cell-derived neurotrophic factor, 20 ng/mL insulin-like growth factor 1, and 1 mM GlutaMAX. NIM also contained small molecules or 1% DMSO in differentiation control groups. From day 0, fibroblasts were treated with NIM containing small molecule combination A (CA: 3 µM CHIR99021, 0.25 µM LDN193189, 10 µM RG108, 1 µM dorsomorphin, 3 µM P7C3-A20, 0.5 µM A83-01, and 5 µM ISX9) for three days. On day 3, half the medium was changed to NIM containing small molecule combination B (CB: 10 µM forskolin, 5 µM Y27632, 1 µM DAPT, 1 µM PD0325901, 0.5 µM A83-01, 1 µM purmorphamine, and 3 µM P7C3-A20) for another 3 days. On day 6, half of the medium was again changed with NIM containing CA and on day 9 with NIM supplemented with CB. On days 10 and 12, neurons were harvested for downstream analyses. A list of small molecules used is included in Supplementary Data 6. A list of fibroblast cell lines included in the generation of induced neurons using small molecules and included in RNA-seq is included in Supplementary Data 7.

## DNA constructs and site-directed mutagenesis

The full-length SETBP1 construct fused to a C-terminal Myc-DDK tag under a human CMV promoter (pCMV-Entry-SETBP1) was purchased from Origene (RC229443). Constructs carrying *SETBP1* variants (pCMV-Entry-SETBP1) were generated using QuikChange Lightning Site-Directed Mutagenesis Kit (Agilent) following the manufacturer's protocol. To generate constructs carrying *SETBP1* fused to a YFP-tag (pYFP-SETBP1), *SETBP1* cDNAs were first subcloned using EcoRI/XhoI restriction sites into a modified pEGFP-C2 vector (Clontech), where the N-terminal EGFP-tag was then replaced with a YFP-tag using BshTI/Bsp1407I restriction sites. To generate AT-rich reporter vectors used in luciferase reporter assay, 100 µM sense and antisense single-stranded oligonucleotides carrying 5′-AAAATAA-3′ or 5′-AAAATAT-3′ repeats were first annealed using 2× annealing buffer [20 mM Tris (Thermo-Fisher), 2 mM EDTA (Sigma), 100 mM NaCl (Sigma), pH 8.0]. Annealed oligonucleotides were phosphorylated using 1× T4 DNA ligase buffer and 30 units of T4 PNK (New England Biolabs), then inserted into a pGL4-luc2 vector using KpnI/HindIII restriction sites. A firefly luciferase reporter construct containing FOXP2 promoters, TSS1 and TSS2, respectively, was a gift from the group of Sonja Vernes[63]. For mammalian one-hybrid (M1H) assays, a pGL4-luc2-GAL4UAS-adenovirus promoter reporter construct and a pGL4.23 construct with an adenovirus major late promoter (pBIND2) were generated as previously described[64]. *SETBP1* cDNAs were subcloned from pCMV-Entry-SETBP1 into the empty pBIND2 using SalI/NotI restriction sites, generating pBIND2-SETBP1 expression vectors. A GFP-SET construct was generated as previously described[6]. All constructs were verified by Sanger sequencing. Primer sequences are listed in Supplementary Data 8.

## Immunofluorescence microscopy

HEK293T/17 cells grown on poly-L-lysine-coated coverslips were transiently transfected with 500 ng of a pCMV-SETBP1 or pYFP-SETBP1 construct. Cells were fixated 48 h after transfection using 4% paraformaldehyde solution (Electron Microscopy Supplies Ltd) for 20 min at room temperature (RT). Human dermal fibroblasts were grown on coverslips coated with 1 µg/ml fibronectin (Corning) and fixated as above. Human induced neurons were fixated as above. Fixated cells were permeabilised with 0.4% Triton-X-100/PBS for 20 min at RT. Following incubation in a blocking solution for 1 h at RT, cells were incubated in primary antibodies overnight at 4 °C and then with conjugated secondary anti-IgG antibodies for 1 h at RT. Hoechst 33342 (Invitrogen) was used for nuclear staining, before mounting with VECTASHIELD® Antifade Mounting Medium (Vectorlab). Fluorescence images were obtained using an LSM880 AxioObserved confocal microscope (Zeiss). For images of single nuclei, the Airyscan unit (Zeiss) was used with a 4.0 zoom factor. A list of antibodies used can be found in Supplementary Data 9.

## Image analysis

Fluorescence images were analysed with ImageJ/FIJI plugins. SETBP1 speckle sizes, Tuj1- and Hoechst-positive neurons were quantified with the ImageJ/FIJI (v1.53t) "Analyse particle" plugin. Percentages of neurons positive for both Tuj1 and Hoechst (neurons) vs Hoechst (all cells) were calculated. Sholl analysis of individual neurons was performed using the ImageJ/FIJI (v1.53e) "SNT" plugin (v4.3.0-pre-release3) using the default settings in the manual.

## Co-immunoprecipitation

HEK293T/17 cells grown in a 10-cm culture dish were transiently transfected with constructs expressing FLAG-SETBP1 only or together with GFP-SET under a human CMV promoter. An empty pCR2.1-TOPO (Invitrogen) was used as a filler to top up to 12 µg DNA in total per transfection. Cells were lysed in 1 ml of Pierce® IP Lysis Buffer (Thermo-Fisher) supplemented with protease inhibitors (Roche) and 1% PMSF

(Thermo-Fisher) 48 h post-transfection. All following steps were performed at 4 °C. Cell lysates were incubated for 10 min under rotation, centrifuged at 16,800×g for 20 min. The supernatant was then quantified with a Pierce BCA protein assay kit (Thermo-Fisher). 5% protein lysate was collected as input and denatured in Laemmli buffer (Bio-Rad). The 400 µg protein was immunoprecipitated with 30 µl FLAG-agarose beads (Thermo-Fisher) or 25 µl GFP-Trap (Chromotek) on a rotating wheel overnight. Beads were then washed three times with PBS and eluted with Laemmli buffer for immunoblotting analysis.

## Immunoblotting and band intensity quantification

HEK293T/17 cells grown in a 6-well culture plate were transiently transfected with 3 µg DNA. Cells were lysed in 1 ml of Pierce® RIPA Buffer (Thermo-Fisher) supplemented with protease inhibitors (Roche) and 1% PMSF (Thermo-Fisher) 48 h post-transfection. Total protein was quantified using a BCA assay. Proteins were resolved on 4–15% Tris-Glycine gels and transferred to PDVF membranes (Bio-Rad). After blotting, membranes were incubated overnight at 4 °C with the appropriate primary antibodies, followed by HRP-conjugated secondary antibodies (Supplementary Data 9). Proteins were visualised using the Novex ECL Chemiluminescent Substrate Reagent kit (Invitrogen) or SuperSignal West Femto Maximum Sensitivity Substrate (Thermo-Fisher) and the ChemiDoc XRS + System (Bio-Rad). Band intensity was quantified using ImageJ. Background-subtracted band intensity was divided by the background-subtracted band intensity of β-actin for normalisation. For quantification of (co-) immunoprecipitation experiments, the background-subtracted band intensity of the immunoprecipitated fraction was normalised with respect to the input fraction for each condition[65].

## Fluorescence-based quantification of protein stability and degradation

HEK293/T17 cells were transfected in triplicate in clear-bottomed black 96-well plates with YFP-tagged SETBP1 variants. After 24 h, cycloheximide (Sigma) at a final concentration of 50 µg/ml was added, MG132 (R&D Systems) at 5 µg/ml and Bafilomycin A1 (R&D Systems) at 100 nM. Cells were incubated at 37 °C with 5% $CO_2$ in the Infinite M200Pro microplate reader (Tecan), and the fluorescence intensity of YFP (Ex: 505 nm, Em: 545 nm) was measured over 24 h at 3-h intervals.

## Luciferase reporter assay and mammalian-one-hybrid (M1H) assay

HEK293/T17 cells were seeded in clear-bottomed white 96-well plates (Greiner Bio-One) and transfected in triplicate using GeneJuice (Merck-Millipore). Cells were co-transfected with 350 ng of firefly luciferase reporter construct containing six repeats of AT-rich consensus sequence (5′-AAAATAA-3′ or 5′-AAAATAT-3′) or FOXP2 promoters (TSS1/TSS2)[63], 6.5 ng of pGL4.74 Renilla luciferase normalisation control, and 700 ng of pYFP construct with or without SETBP1. For the M1H assay, cells were co-transfected with 1433 ng (50 nM) of expression construct containing only GAL4 or GAL4 including SETBP1, 416 ng of the reporter construct with or without GAL4-binding site, and 165 ng of Renilla luciferase normalisation control. After 48 h, firefly luciferase and Renilla luciferase activities were measured using a Dual-Luciferase Reporter Assay system (Promega) according to the manufacturer's instructions at an Infinite M Plex Microplate reader (Tecan).

## Cell proliferation assay

Fibroblasts were seeded in triplicate in clear-bottomed black 96-well plates (Greiner Bio-One). Cell proliferation was measured using a CyQUANT™ Direct Cell Proliferation Assay (Thermo-Fisher) according to the manufacturer's instructions at an Infinite M Plex Microplate reader (Tecan). Fluorescence intensity was measured daily for four days. Background fluorescence was subtracted, and values were normalised to day 0. Growth curves were plotted as fluorescence vs time.

## RNA sequencing (RNA-seq) and data analysis

Three technical replicates of each fibroblast cell line and three biological replicates of induced neuron differentiation were included in the RNA-seq experiments. A list of cell lines included can be found in Supplementary Data 5. Total RNA was extracted from one to two million cells. 1 µg of total RNA extracted (Qiagen) was used to generate RNA libraries using NEBNext® UltraTM RNA Library Prep Kit for Illumina® (New England Biolabs) following the manufacturer's protocol, and index codes were added to attribute sequences to each sample. The libraries were sequenced on a NovaSeq6000 platform using 150 bp pair-end reads by Novogene. Image processing and basecalling were performed using the Illumina Real Time Analysis Software. Fastq files were aligned to the human genome (GRCh38/hg39, Ensembl) using HISAT2[66] software together with the corresponding splice junctions Ensembl GTF annotation. At least 89% of the reads for each sample were uniquely mapped to the human genome, entailing at least 36.4 million unique reads in the sample with the lowest sequencing depth for fibroblast samples. For induced neuron samples, at least 88% of the reads for each sample were uniquely mapped to the human genome, entailing at least 39.9 million unique reads in the sample with the lowest sequencing depth. Data pre-processing and analysis were performed using R Statistical Software v4.1.2[67]. The count table was created using featureCounts in Rsubread[68]. Transcripts that have no reads in all samples were removed. Read counts were then normalised to transcripts per million (TPM) within each sample and log-transformed. For fibroblast samples, genes expressed at TPM < 1 in ten samples or more were then filtered out. This resulted in a list of 13,116 genes. For induced neuron samples, genes expressed at TPM < 1 in more than 10% of the samples were filtered out. This resulted in a list of 18,639 genes. Unbiased clustering of the samples with principal component analysis (PCA) and hierarchical analysis was performed on variance stabilising transformed data using the 500 most variable genes. Batch correction was done using limma[69] (v.3.50.3) for data visualisation. Outlier samples were removed. Differential gene expression of fibroblasts was performed using DESeq2[70] (v.1.34.0) with design: -sex + batch + age_at_sampling + condition [$p < 0.05$, false discovery rate (FDR)], accounting for the variables: sex, batch and age at sampling. Differential gene expression of induced neurons was performed using DESeq2[70] with design: design2 = -sex + batch + day + condition [$p < 0.05$, false discovery rate (FDR)], accounting for the variables: sex, batch and differentiation day in culture. Differentially expressed genes (DEGs) were filtered for an adjusted $p$-value < 0.1 (FDR), and those with a $\log_2$ fold change larger than 1 or smaller than −1 were retained for further analyses. Gene ontology over-representation analysis was performed using the topGO R-package[39] (v.2.46.0). A list of expressed genes in our fibroblast or induced neuron samples was used as the custom background for gene ontology analysis with multiple testing correction ($p < 0.05$, Benjamini-Hochberg FDR). In order to reduce redundancy of the identified GO terms, all significant (adjusted $p$-value < 0.05) terms were used as input to REVIGO[71] http://revigo.irb.hr/ or the rrvgo R-package (v.1.6.0). The value of the resulting gene list of 0.5 was used. The resulting non-redundant GO processes were reported. Degrees of overlap between the differentially expressed genes in our fibroblast samples and autism (v.0.22)- or intellectual disability (ID, v.3.2)-associated genes in the PanelApp database were calculated using a Fisher's exact test ($p < 0.05$) in R. Degrees of overlap between the putative SETBP1 binding targets identified in a previously published ChIP-seq experiment[22] and autism (v.0.22)- or intellectual disability (ID, v.3.2)-associated genes in the PanelApp database were calculated using a Fisher's exact test ($p < 0.05$). A list of expressed genes in our fibroblast or induced neuron samples was used as the custom background. Venn diagrams were made using https://bioinformatics.psb.ugent.be/webtools/Venn/ or the VennDiagram (v.1.7.3) R-package. Rank–rank hypergeometric overlap analysis was performed to identify overlap between gene-

expression signatures using the default settings of the RRHO (v.1.42.0) R-package[40,41]. Code used in the RNA-seq analysis is available here: https://github.com/galagoz/setbp1-missense.

## Quantitative real-time PCR

Total RNA was extracted with Qiazol or an RNeasy Plus mini kit (Qiagen) following the manufacturer's protocols. 1 μg of total RNA was used to synthesise cDNA using SuperScript III Reverse Transcriptase (Thermo-Fisher). Real-time PCR was performed using iQ SYBR Green Supermix (Bio-Rad) at a CFX384 Touch Real-Time PCR Detection System (Bio-Rad).

## Statistical analysis of cell-based functional assays

Statistical analyses for cell-based functional assays were done with GraphPad Prism (version 8.0) Software. All experiments were performed at least three times, and the performed numbers are provided in the respective Figure legends. All statistical tests used are detailed in the corresponding Figure legends. Two-tailed Student's $t$-test was used to evaluate the significance of mean differences between the control and the patient groups. Differences between groups were evaluated using a one- or two-way ANOVA followed by a Bonferroni, Tukey, or Dunnett $post\text{-}hoc$ test. A $p$-value < 0.05 was considered significant. Exact $p$-values are shown inthe Figures. Source data with detailed statistical tests and results are included in the Source Data Files.

## Tools

These tools have been used in this study:

CADD: https://cadd.gs.washington.edu
COSMIC: https://cancer.sanger.ac.uk/cosmic
DECIPHER: https://www.deciphergenomics.org/gene/SETBP1/
gnomAD: https://gnomad.broadinstitute.org
MetaDome: https://stuart.radboudumc.nl/metadome
MTR-Viewer: https://biosig.lab.uq.edu.au/mtr-viewer/
OMIM: https://www.omim.org
Spatial clustering: https://github.com/laurensvdwiel/SpatialClustering
SNT: https://imagej.net/plugins/snt/
UniProt: https://www.uniprot.org
PhenoScore: https://github.com/ldingemans/PhenoScore.

## Reporting summary

Further information on research design is available in the Nature Portfolio Reporting Summary linked to this article.

## Data availability

The raw RNA sequencing data contain the genetic information from a rare disease population and are therefore protected by European data privacy laws [European General Data Protection Regulation (GDPR)] and not publicly available. Requests for access to the raw RNA sequencing data should be addressed to the corresponding authors, subject to a data use agreement. The intermediary RNA sequencing data (.bam files) generated and analysed in this study have been deposited in the permanent public data archive of the Max Planck Institute (Nijmegen) database under a permanent accession code https://hdl.handle.net/1839/69419e36-6b6a-4e2d-9ebb-c1f4bbca6c2f. These intermediary RNA sequencing data are available under restricted access because they contain the genetic information from a rare disease population and are therefore protected by the European data privacy laws [European General Data Protection Regulation (GDPR)], and can only be stored in the permanent public data archive of the Max Planck Institute (Nijmegen) database with restricted access, as approved by the ethics committee. These intermediary data can be downloaded by academic users with institution email addresses upon request. To access these intermediary data, after creating a user account on the public data archive of the Max Planck Institute website https://archive.mpi.nl/mpi/user, academic users are asked to submit a

request following detailed instructions on the website via this form: https://archive.mpi.nl/mpi/contact (processing time is within three months and depends on the completeness of information provided by the requesters). Following the approval of the request, a material transfer agreement will be arranged before the release of the intermediary data. The processed RNA sequencing data are available via GEO accession numbers: https://www.ncbi.nlm.nih.gov/geo/query/acc.cgi?acc=GSE301238 (fibroblasts) and https://www.ncbi.nlm.nih.gov/geo/query/acc.cgi?acc=GSE301239 (induced neurons). No gene panel, Sanger or next-generation DNA sequencing data for identifying patients were generated in this study. Previously reported variants are indicated in Supplementary Data 1. Patients identified in a diagnostic context were not recruited for this study; consequently, no DNA sequencing was performed, and no sequencing data are available for deposition. Only clinical and genotype information were obtained. The clinical and speech phenotype data are provided in the Supplementary Data 1 and 2 and Source Data files. Further enquiries regarding genetic, clinical and speech information should be directed to the corresponding authors. Source data are provided with this paper.

## Code availability

We did not generate any original code in this study.

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

## Acknowledgements

We are grateful to all individuals and families for their contributions. We would like to thank the members of the Cell Culture Facility, Department of Human Genetics, Radboud University Medical Centre, Nijmegen, for cell culture of proband-derived cell lines. We especially thank Else Eising (Language and Genetics Department, MPI for Psycholinguistics, Nijmegen), Alexander Hoischen, Lot Snijders-Blok (Department of Human Genetics, Radboud University Medical Centre, Nijmegen), and Rocio Acuna-Hidalgo (Nostos Genomics, Berlin) for scientific discussion. This work was supported by the Max Planck Society (M.M.K.W., R.A.K., G.A., J.C., J.d.H., W.J.J.C., A.V., and S.E.F.). W.K.C. was funded by the SFARI and the JPB Foundation. M.S.H., I.E.S., and A.T.M. were funded by the National Health and Medical Research Council (NHMRC) Centre of Research Excellence Grant (1116976), the Australian Research Council (ARC) Discovery Project (DP120100285), the NHMRC Project Grant (1127144), and the March of Dimes Grant Scheme. M.S.H. was funded by the NHMRC Career Development Fellowship (1063799). I.E.S. was funded by the NHMRC Investigator Grant (1172897), the NHMRC Practitioner Fellowship (1006110), and the NHMRC Development Grant (1153614). A.T.M. was funded by the NHMRC Investigator Grant (1195955), the NHMRC Practitioner Fellowship (1105008), and the NHMRC Development Grant (1153614). E.P. was supported by the National Health and Medical Research Council (GNT11149630), Australia and the research WGS was supported by NHMRC. E.M. was supported by the FU Starting Grant, Nr. 43 mit FK-Beschluss vom490 06.02.2023. B.W.M.vB. is a member of the European Reference Network on Rare Congenital Malformations and Rare Intellectual Disability ERN-ITHACA. ERN-ITHACA is funded by the European Union, under the grant agreement N°101156387.

## Author contributions

M.M.K.W. and S.E.F. conceived the idea. M.M.K.W., A.T.M., B.W.M.vB., and S.E.F. designed research. M.M.K.W., R.A.K., R.O.B., G.A., A.J.M.D., J.C., J.d.H., E.M., W.J.J.C. and A.V. performed experiments. M.M.K.W., R.A.K., R.O.B., G.A., A.J.M.D. and J.C. analysed data. M.S.H., C.B., M.B., A.B., D.C., F.E., G.B.F., I.M.B.H.vdL, A.M., D.M., L.M., F.N., A.P., I.E.S., F.S., S.J.T., A.V., S.W., W.K.C., M.G., V.L.G., E.P., A.T.M. and B.W.M.vB. were responsible for the clinical and genetic analysis of patients. M.M.K.W. wrote the initial draft of the manuscript. M.M.K.W., A.T.M., B.W.M.v.B., and S.E.F. supervised research. All authors reviewed the manuscript.

## Funding

## Competing interests

The authors declare no competing interests.

## Additional information

Maggie M. K. Wong [1] ✉, Rosalie A. Kampen[1], Ruth O. Braden[2,3], Gökberk Alagöz [1], Michael S. Hildebrand [2,4], Alexander J. M. Dingemans[5], Jean Corbally[1], Joery den Hoed [1], Ezequiel Mendoza [6], Willemijn J. J. Claassen[1], Christopher Barnett[7], Meghan Barnett[7], Alfredo Brusco [8,9], Diana Carli [10,11], Bert B. A. de Vries [5], Frances Elmslie[12], Giovanni Battista Ferrero[13], Nadieh A. Jansen[5], Ingrid M. B. H. van de Laar[14], Alice Moroni [8], David Mowat[15,16], Lucinda Murray[17], Francesca Novara[18], Angela Peron [19,20,21], Ingrid E. Scheffer [2], Fabio Sirchia[22], Samantha J. Turner[2,3], Aglaia Vignoli[23], Arianna Vino[1], Sacha Weber[24], Wendy K. Chung[25,26], Marion Gerard[24], Vanesa López-González[27], Elizabeth Palmer [16,17], Angela T. Morgan [3,28], Bregje W. van Bon[5] & Simon E. Fisher [1,29] ✉

[1]Language and Genetics Department, Max Planck Institute for Psycholinguistics, Nijmegen, The Netherlands. [2]Epilepsy Research Centre, Department of Medicine and Paediatrics, University of Melbourne Austin Health Victoria, Melbourne, VIC, Australia. [3]Speech and Language, Murdoch Children's Research Institute, Melbourne, VIC, Australia. [4]Neuroscience Group, Murdoch Children's Research Institute, Royal Children's Hospital, Melbourne, VIC, Australia. [5]Department of Human Genetics, Radboud University Medical Center, Nijmegen, The Netherlands. [6]Institut für Verhaltensbiologie, Freie Universität Berlin, Berlin, Germany. [7]SA Clinical Genetics Service, Women's and Children's Hospital, North Adelaide, SA, Australia. [8]Department of Neurosciences Rita Levi-Montalcini, University of Torino, Torino, Italy. [9]Medical Genetics Unit, Città della Salute e della Scienza, University Hospital, Turin, Italy. [10]Department of Medical Genetics, University of Torino, Torino, Italy. [11]Pediatric Onco-Hematology, Stem Cell Transplantation and Cell Therapy Division, Regina Margherita Children's Hospital, Città della Salute e della Scienza di Torino, Torino, Italy. [12]South West Thames Regional Genetics Service, St George's, University of London, London, UK. [13]Department of Clinical and Biological Sciences, University of Torino, Orbassano (Torino), Italy. [14]Department of Clinical Genetics, Erasmus MC, University Medical Center Rotterdam, Rotterdam, The Netherlands. [15]Centre for Clinical Genetics, Sydney Children's Hospitals Network Randwick, Randwick, NSW, Australia. [16]Discipline of Paediatrics and Child Health, Faculty of Clinical Medicine, UNSW, Randwick, NSW, Australia. [17]Genetics of Learning Disability Service, Royal North Shore Hospital, St Leonards, Sydney, NSW, Australia. [18]Microgenomics srl, Pavia, Italy. [19]Human Pathology, ASST Santi Paolo e Carlo, San Paolo Hospital, Milan, Italy. [20]Child Neuropsychiatry Unit—Epilepsy Center, ASST Santi Paolo e Carlo, San Paolo Hospital, Department of Health Sciences, Università degli Studi di Milano, Milan, Italy. [21]Division of Medical Genetics, Department of Pediatrics, University of Utah, Salt Lake City, UT, USA. [22]Department of Molecular Medicine, University of Pavia, Pavia, Italy. [23]Child Neuropsychiatry Unit, ASST Grande Ospedale Metropolitano Niguarda, Department of Health Sciences, Università degli Studi di Milano, Milan, Italy. [24]Department of Genetics, Reference center for Rare Diseases and Developmental Anomalies, Caen, France. [25]Department of Pediatrics, Boston Children's Hospital Boston, Boston, MA, USA. [26]Harvard Medical School, Boston, MA, USA. [27]Sección de Genética Médica, Servicio de Pediatria, Hospital Clinico Universitario Virgen de la Arrixaca, IMIB-Arrixaca, CIBERER-ISCIII, Murcia, Spain. [28]Department of Audiology and Speech Pathology, University of Melbourne, Melbourne, VIC, Australia. [29]Donders Institute for Brain, Cognition and Behaviour, Radboud University, Nijmegen, The Netherlands. ✉e-mail: maggie.wong@mpi.nl; simon.fisher@mpi.nl

