## [Transparent Peer Review file · Nature Communications]

SETBP1 variants outside the degron disrupt DNA-binding, transcription and neuronal differentiation capacity to cause a heterogeneous neurodevelopmental disorder

Corresponding Author: Dr Maggie Wong

Version 0:

Reviewer comments:

Reviewer #1

(Remarks to the Author)

In this manuscript, entitled “SETBP1 variants outside the degron disrupt DNA-binding and transcription to cause a heterogeneous neurodevelopmental disorder,” Wong and colleagues present a thorough and intriguing series of clinical and molecular assessments aimed at identifying the functional impact of previously uncharacterized SETBP1 variants in human patients. Previously, a number of germline de novo SETBP1 variants have been shown to cause clinically distinct and heterogeneous neurodevelopmental disorders – e.g., Schinzel-Giedion syndrome (caused by missense variants at an important degron region) and SETBP1-haploinsufficiency disorder. However, owing to a lack of comprehensive genotype-phenotype association studies, the field’s limited understanding of roles for SETBP1 during neurodevelopment and the extent of clinical heterogeneity observed among patients harboring SETBP1 variants, more work is needed to fully understand the contribution of these variants to clinical outcomes. Here, the authors present a comprehensive investigation of the largest cohort to-date (N=18) of individuals carrying SETBP1 missense variants outside the degron, where they performed both clinical and speech phenotyping, along with functional studies in cells to explore contributions of these variants to cellular and transcriptomic phenotypes. For example, they found that these variants result in clinically and functionally variable developmental syndromes (displaying only partial overlaps with classical SGS and SETBP1-haploinsufficiency) and provide evidence that these variants appear to result in loss-of-function pathophysiological mechanisms associated with impaired protein degradation (via deficits in ubiquitination), DNA binding and transcription in patient fibroblasts and HEK293 cells transfected with SETBP1 variants. Interestingly, the impact of these variants appears to be independent of overall protein abundance, which is in contrast to SGS and SETBP1 haploinsufficiency. Overall, this is an interesting and well controlled study, which nicely weaves together important clinical phenotyping assessments of non-degron associated SETBP1 missense variants (using a large cohort of patients) with functional assays in cells to further explore their mechanistic relevance (an approach that should be applauded). However, in certain circumstances, it remains unclear how the cellular assays employed (particularly in the case of the fibroblast proliferation assays) may relate to the neurological phenotypes observed in patients. In addition, some of the findings described in this manuscript appear to be overstated based upon the evidence provided.

1. The fibroblast proliferation assays, while interesting (i.e., greater proliferation with non-degron associated variants in SETBP1), seem to provide little insight into how these variants may affect post-mitotic neurons (assuming that neurons are the cell-types primarily affected by SETBP1 mutations in patients – is this known?). Using hPSC-derived neurons, it would be helpful to compare the phenotypic effects of expressing WT vs. degron associated vs. non-degron associated mutants of SETBP1 in order to verify that these different mutant types have distinct functions in a relevant human cell type. Even just comparing one mutant of each type (vs. WT SETBP1) to validate the transcriptional effects predicted from the fibroblast studies in neurons would greatly strengthen this manuscript (although further neuronal phenotyping would also be a welcome addition).
2. In Figure 2, the authors make the argument that all variants of SETBP1 (degron associated vs. non-degron associated) lead to increased abundance of the protein in patient derived fibroblasts but do not affect interactions with SET or pPP2A/PP2A ratios. However, in Figure 2A, there appears to be quite a bit of variability in the expression of B-actin (their loading control), which may influence interpretations of these data. It would be helpful to run all samples (Controls vs. variants) on the same blots and ensure equal loading. These data could then be quantified (Control vs. degron associated variants vs. non-degron associated variants) and stats applied.

3. In Figure 2C, they make the point that exogenously expressed SETBP1 variants (vs. WT) display elevated abundance, which is quantified in Figure 2D. However, it does not appear that they validated equal expression of the exogenously expressed transcripts to ensure that the variants are simply not more highly transcribed vs. WT (unless I missed this). This should be performed. Also, while it appears true that the conglomerate effect of the variants (vs. WT) do not impact SET interactions or pPPP2A/PP2A levels, it seems that some of the variants may indeed affect these phenomena (e.g., E858K vs. control seems to lead to increased pPPP2A). Is this just an issue of variability in sample quantifications, or do the authors believe that some of the variants may indeed have differential effects?

4. The observation that “fibroblasts carrying SETBP1 variants outside the degron showed distinct transcriptomic profiles from healthy control, SGS and SETBP1-haploinsufficiency disorder patients,” while true to some degree, seems overstated. In fact, from the data in Figure 5, it appears that much fewer gene expression differences were observed for variants outside the degron domain vs. within the degron domain, and many of these changes (vs. healthy controls) are similar between the different variants. As such, it seems equally plausible that these overlapping genes might also be important for driving the phenotypic effects in patients. One suggestion might be to perform RRHO analyses to compare the impact of these different variants vs. healthy controls to get a broader sense of how similar/different they may be.

5. Along the lines of #4 above, the authors state that from a transcriptional standpoint that these different variant classes cluster independent (Figure 5B), however, this is overstated, as most of the variant classes appear to cluster together. In addition, there appears to be a tremendous amount of variability among the control samples, so I wonder if the authors could comment on this further.

In sum, this is an interesting paper that provides much needed insights into the functional consequences of novel SETBP1 variants that result in developmental syndromes. However, numerous issues related to the manner in which the functional data are being interpreted need to be addressed prior to publication.

Reviewer #2

(Remarks to the Author)

Wong and colleagues have provided a great description of not only the known SETBP1 variants within the degron or causing haploinsufficiency, but also this novel group of missense variants. In general, the manuscript is well written and easy to follow. I only have a few edits and questions.

I am glad you brought up that iPSCs could be used. My first thoughts were “wow this would be great in neural progenitor cells!” But then bringing up that the SGS models were not as helpful as expected is interesting. While certainly cost prohibitive, it is interesting to consider that some of these neurodevelopmental disorder won't work with that model. Are there other disorders that have had similar issues? (of course, they might not be published, so that may be an impossible question!)

For your functional studies, should the controls also be age (or approximately age) matched? I'm concerned older individuals may have slower proliferation anyway. You could even use other affected individuals' (just not SETBP1 affected) cells, or try to correct the patient cells with CRISPR (although that's one of the things that might be easier in iPSCs than fibroblasts). Also, a chart (like from GTEx) showing the different isoforms in different cells could be informative.

I think you touched on this, but is there a clear distinction between the DEGs and the disorders? Looks like Wnt signaling is impaired for variants w/in the degron, could that be partially explanatory of some of the more physical phenotypes observed? You highlight FOXP2, but I think some others could be highlighted as well.

You suggest that clinicians use available tools to assess variants. Do you have a suggestion for a protocol for this? You suggest the ion channels tool, which is great, but I think if you could say pathogenic variants have greater than X CADD score or some other tools, that would be great. You have a lot of tools (CADD, SIFT, PolyPhen, MetaDome) but you could also add MTR.

Are there any SETBP1 duplications? I wonder if that increased protein would be problematic or if the increased protein plus a variant is needed?

Finally, I think there is an opportunity for a helpful figure, something that connects all 3 SETBP1-related disorders. Maybe a spectrum along the protein or a chart? There is a lot of information on the three disorders, so having a figure and/or table would be helpful. Kind of like Figure 6, but for phenotypes or HPO terms.

A few edits:

Abstract:

Line 70-

This sentence in the abstract is too long and hard to read: “However, due to the lack of systematic investigation of genotype-phenotype associations of different types of SETBP1 variants, and limited understanding of its roles in neurodevelopment, the extent of clinical heterogeneity and how this relates to underlying pathophysiological mechanisms remain elusive, imposing challenges for diagnosis.”

Line 75-

I would say “degron region” since you haven't had a chance to define it yet

Results:
Line 208-
What is MSH?

Line 220-
I think midface retrusion is more commonly used

Line 225 –
Move the “(amino acids 854-858)” to after “within the SKI domain...”

Line 310-
This is explaining your strategy, but does not give the results, maybe change the formatting?

Overall, this is a really nice paper explaining a complicated set of disorders!

Reviewer #3

(Remarks to the Author)

The study conducted by Wong and colleagues aims to illustrate the heterogeneity in molecular and clinical characteristics of SETBP1 variants through the use of in vitro functional assays and transcriptomic analyses. Interestingly, the authors could show that patients' phenotypical expression varies and seems to be influenced by the proximity to the degron region. Therefore, the authors selected and tested various variants in their subsequent experiments, differentiating and considering their distances between those located within and outside of the canonical degron.

Wong and colleagues show that a subset of SETBP1 variants outside the degron leads to an increase in protein levels and promotes cell proliferation. Moreover, SETBP1 degradation via autophagy or proteasomal pathways was impacted, but the degree and direction of degradation varied depending on the variant's distance from the degron region. Similarly, a subset of variants showed reduced transcriptional activity.

Lastly, the authors performed RNA-seq in patient-derived fibroblasts, showing distinct differential expression patterns depending on SETBP1 variants.

The study represents a valuable resource for clinical diagnostic and mechanistic research. In addition, the study represents a very nice example of the degree of heterogeneity of phenotypes, even when different variants are studied side by side. To this extent, Figure 6 is very impactful.

However, several aspects limit the impact of this study and should be considered to improve the manuscript.

- 1) The role of the observed phenotypes DNA-binding/transcription in the emergence of the clinical presentation remains elusive. This is a critical limitation of the manuscript. The authors should either develop this aspect further (see also comments on the cell type used) or remove some (over)statements and highlight this limitation throughout the manuscript.
- 2) The authors claim that they have identified FOXP2 as a direct transcriptional target of SETBP1 through their cell-based experiments. However, they have not provided any direct evidence to support this conclusion, leaving the reasoning behind it unclear. In line with that, the results depicted in Figure 4C show reporter constructs for FOXP2 promoters without giving a rationale for choosing this transcriptional target. Furthermore, the authors generated RNA-seq data from multiple patient fibroblasts that could be utilized to confirm the transcriptional regulation of their target FOXP2. Is there any indication within the RNA-seq dataset that suggests FOXP2 is differentially regulated in mutant cells? The authors should comment on these findings.
- 3) According to the authors, variants located farthest from the degron (specifically, p.(Ser444Arg) and p.(Val657Ala)) exhibit impaired SETBP1 degradation due to disrupted ubiquitination. However, the data presented in Figure 3 only partially support these findings, as the ubiquitination levels of both variants are similar to those of wild-type SETBP1. To avoid misinterpretation of results, the western blot (Fig. 3C) should be repeated since the input of wild-type SETBP1 appears to be at lower levels compared to the other variants, as indicated by the reduced intensity of the loading control β -actin.
- 4) The authors have emphasized at different points that the causes and outcomes related to SETBP1 variants are likely to be specific to particular cell types. However, the current manuscript employs different cell types. Hence, it is necessary for the authors to address how this may affect their results.
- 5) Related to the point above, the authors in several points claim that their work addresses the pathophysiological mechanisms of SETBP1 mutations. However, as patients with mutations in this gene have CNS issues, more work on CNS-relevant cells/models would better support this claim. This is one of the major limitations of this study.
- 6) According to Wong et al., SETBP1 variants impact multiple protein functions. While functional assays and western blots are valuable for validating their hypotheses, proteomic analyses would have been a more suitable option. This is particularly relevant since the authors noted differences in protein degradation, which could have allowed for the identification of direct consequences on protein levels and potential interactions with SETBP1. Additionally, these findings could have been cross-referenced with RNA-seq data to explore direct SETBP1 interactors.
- 7) The authors investigate transcriptomic profiles from different SETBP1 variants, pointing out that the various modalities (outside, within, truncating) are largely separated clusters (ad Fig. 6B). However, as indicated in Fig 6B, the variants are actually intermingled, showing an overlap throughout. The authors should revise and comment on that.

Minor comments:

1) The figure panels are relatively compact, making it challenging to differentiate between individual data markers, particularly in Fig. 2 and 3.

2) The labeling in Figure 3C,D is incorrect as the variant p.(Val657Ala) is erroneously abbreviated as "V647A". The authors should correct this error.

3) Ad line 371: "to inhibit" not "to inhibition"

4) ad line 691: The sentence contains a fraction written in italics that should be changed to a regular font.

5) ad line 707: The sentence abruptly concludes with the phrase "in SETBP1-related disorders and. To study..."; this part needs to be revised to improve clarity and comprehension.

Version 1:

Reviewer comments:

Reviewer #1

(Remarks to the Author)

In this substantially revised manuscript, the authors have done a commendable job in responding to my previous critiques, as well as those of my fellow Reviewers, both through the addition of new data (including from iPSC neurons, as requested) and text revisions. I now feel that this manuscript is suitable for publication at Nature Communications and will be of broad interest to the field.

Reviewer #2

(Remarks to the Author)

The authors nicely describe SETBP1-related disorders and how they vary by variant type and location, particularly the missense variants outside the degron region. Having multiple variables (variant type, location, and phenotype) leads to complex analyses. The clinical phenotyping and molecular studies support their conclusions well. These findings expand the known features of SETBP1-related disorders and provide molecular studies to be used to identify therapeutics.

This work is very significant to the SETBP1-related disorders field. Additionally, their approach will likely benefit other neurodevelopmental disorders with varying missense variants and hotspots. They also use an interesting approach with correlating the phenotypes, facial features, and variant location that I think will be relevant to other disorders.

The in depth molecular phenotyping provides a wealth of novel information. Being able to compare the different variant types across multiple cell types is very valuable. This is a very nice manuscript that provides very important information.

Reviewer #3

(Remarks to the Author)

The authors addressed all my criticisms and revised the manuscript to recognize the limitations in the experimental design and model systems used. The overall organization of the manuscript has improved. The authors have now utilized induced neurons for transcriptomics, significantly improving the quality and insights regarding SETBP1 variants.

I suggest thoroughly checking the language and figure references, as some mistakes remain. Below are some examples I have noted.

Line 274: "...and to aid diagnosis."

Line 394: refer to Suppl. Fig. 4a not b

Line 399: refer to Suppl. Fig. 4b not c

Line 405: refer to Suppl. Fig. 4c (else there is no reference to this panel in the whole text)

Line 421: there is no Suppl. Fig. 4i

Line 427: there is no Suppl. Fig. 4j

Line 563: refer to Suppl. Fig. 10b-f

Line 609: ("days 10 and12") put a space before 12

Line 939: there is no Suppl. Figure 5d

Response to reviewer comments

We would like to thank the reviewers for their positive feedback and constructive comments on our manuscript, which we found very helpful. The revision process took significantly longer than expected since the leading author has unfortunately had a serious long-term sickness which inevitably delayed the progress of the additional experiments and analyses. Yet, through her substantial efforts, we have been able to finish the revised manuscript and are confident that we have satisfactorily addressed all the issues raised. We are extremely grateful to the editors for their understanding and patience under these unusual circumstances, and for the opportunity to resubmit. Our detailed point-by-point responses to each reviewer comment are given below in blue. Quoted newly added text from our main manuscript are “**in blue + bold**”.

Reviewer #1 (Remarks to the Author):

In this manuscript, entitled “SETBP1 variants outside the degron disrupt DNA-binding and transcription to cause a heterogenous neurodevelopmental disorder,” Wong and colleagues present a thorough and intriguing series of clinical and molecular assessments aimed at identifying the functional impact of previously uncharacterized SETBP1 variants in human patients. Previously, a number of germline de novo SETBP1 variants have been shown to cause clinically distinct and heterogenous neurodevelopmental disorders – e.g., Schinzel-Giedion syndrome (caused by missense variants at an important degron region) and SETBP1-haploinsufficiency disorder. However, owing to a lack of comprehensive genotype-phenotype association studies, the field’s limited understanding of roles for SETBP1 during neurodevelopment and the extent of clinical heterogeneity observed among patients harboring SETBP1 variants, more work is needed to fully understand the contribution of these variants to clinical outcomes. Here, the authors present a comprehensive investigation of the largest cohort to-date (N=18) of individuals carrying SETBP1 missense variants outside the degron, where they performed both clinical and speech phenotyping, along with functional studies in cells to explore contributions of these variants to cellular and transcriptomic phenotypes. For example, they found that these variants result in clinically and functionally variable developmental syndromes (displaying only partial overlaps with classical SGS and SETBP1-haploinsufficiency) and provide evidence that these variants appear to result in loss-of-function pathophysiological mechanisms associated with impaired protein degradation (via deficits in ubiquitination), DNA binding and transcription in patient fibroblasts and HEK293 cells transfected with SETBP1 variants.

Interestingly, the impact of these variants appears to be independent of overall protein abundance, which is in contrast to SGS and SETBP1 haploinsufficiency. Overall, this is an interesting and well controlled study, which nicely weaves together important clinical phenotyping assessments of non-degron associated SETBP1 missense variants (using a large cohort of patients) with functional assays in cells to further explore their mechanistic relevance (an approach that should be applauded).

We are grateful to the reviewer for this positive assessment and for recognizing the value of our integrated approach to bring together clinical phenotyping and cell-based functional assays.

However, in certain circumstances, it remains unclear how the cellular assays employed (particularly in the case of the fibroblast proliferation assays) may relate to the neurological phenotypes observed in patients. In addition, some of the findings described in this manuscript appear to be overstated based upon the evidence provided.

1. The fibroblast proliferation assays, while interesting (i.e., greater proliferation with non-degron associated variants in SETBP1), seem to provide little insight into how these variants may affect post-mitotic neurons (assuming that neurons are the cell-types primarily affected by SETBP1 mutations in patients – is this known?). Using hPSC-derived neurons, it would be helpful to compare the phenotypic effects of expressing WT vs. degron associated vs. non-degron associated mutants of SETBP1 in order to verify that these different mutant types have distinct functions in a relevant human cell type. Even just comparing one mutant of each type (vs. WT SETBP1) to validate the transcriptional effects predicted from the fibroblast studies in neurons would greatly strengthen this manuscript (although further neuronal phenotyping would also be a welcome addition).

We thank the reviewer for the suggestion to also investigate functional impacts of variants in a more CNS-relevant model. There are thus far three studies that have studied the impacts of classical SGS variants and *SETBP1*-knockout in human CNS-relevant models¹⁻³. Results from these studies suggested that defects could already be detected in neural progenitors (NPCs) and that their proliferation was impaired. The investigation of variant impacts on different subtypes of neurons requires a different study design, long-term culture, and more detailed investigation. This would be a promising follow-up study, and aligns with ongoing efforts in our research group, but is beyond the scope of the current manuscript. Nonetheless, we do recognise the importance of showing that different groups of variants cause defects in neural cells. We therefore performed an additional substantial new set of experiments testing the impacts of two variants from each

group (classical SGS, missense variants outside degron and truncating variants) in human induced neurons derived from patient fibroblasts.

The paper has been updated with additional sections in the Methods: **“Generation of induced neurons from fibroblasts using small molecules”** (lines 949-968) and Results: **“*SETBP1* variants cause differential effects on differentiation capacity, morphologies, and transcriptomic profiles in induced neurons”** (lines 537-595) and extra Figures showing our findings (Fig 6 and Supplementary Fig 11-14).

We have updated with additional paragraphs in the Discussion that summarised our findings: **“Building on the findings from our broader screen, we showed that the differentiation capacity, morphologies, and transcriptomic profiles of induced neurons derived from patient cells differ from controls, uncovering differential effects for different variant groups, along with variability within groups.”** (lines 632-635)

Moreover, explanations of the choice of neuronal models are added to the Discussion:

“Although patient fibroblasts are not the most severely affected cell type, they are relatively easy to assess and endogenously express *SETBP1*. Since our study placed a strong emphasis on using cell-based functional assays and transcriptomics to screen for pathogenicity and improve definition of *SETBP1*-related disorders, we chose cellular models that allow for systematic functional screening of variants. Investigation of the spatial and temporal expression of *SETBP1* and the functional impact of variants during brain development are much needed to understand the pathways that go awry. This could be achieved by using neuronal models that carry *SETBP1* variants and endogenously express *SETBP1*. To date, only a handful of studies have investigated the impacts of an SGS variant or CRISPR-engineered *SETBP1* deletions or truncating variants in early stage human neural models¹⁻³. However, the impact of *SETBP1* missense or truncating variants have not been studied in neurons and there has not been any direct comparison across groups of variants. Models derived from iPSCs are not suitable for functional screening of variants since the generation of cells carrying selected variants either from patient cells or introduced using gene-editing prior to long-term culture of differentiated cells require a different study design, extensive quality control and more detailed investigation, which is labour-intensive. In this study, to systematically compare the neuronal impacts across variant groups, we studied a smaller selection of representative variants using a method

that enables relatively rapid generation of induced neurons directly from patient fibroblasts using small molecules. Notably, we showed that the differentiation capacity, morphologies, and transcriptomic profiles of induced neurons derived from patient cells differ from controls, uncovering differential effects for different variant groups, along with variability within groups. We note that the differentiation efficiency and the neuronal subtypes obtained using this method is still relatively limited. Future work comparing the functional impacts of different types of variants in models comprising different neuronal subtypes from different brain regions is essential to further delineate the aetiology, complexity and pleiotropy of SETBP1-related disorders. Importantly, the present study provides valuable insights into which variants should be prioritized for labour-intensive follow-up work involving a subset of variants, chosen based on the results obtained from our functional assays.” (lines 813-842)

2. In Figure 2, the authors make the argument that all variants of SETBP1 (degron associated vs. non-degron associated) lead to increased abundance of the protein in patient derived fibroblasts but do not affect interactions with SET or pPP2A/PP2A ratios. However, in Figure 2A, there appears to be quite a bit of variability in the expression of B-actin (their loading control), which may influence interpretations of these data. It would be helpful to run all samples (Controls vs. variants) on the same blots and ensure equal loading. These data could then be quantified (Control vs. degron associated variants vs. non-degron associated variants) and stats applied.

Fig 2A of our original submitted manuscript showed that although the actin loading controls of the patient human dermal fibroblasts (HDFs) were less than the control HDFs, the patient HDFs still showed higher SETBP1 levels, suggesting that SGS variants lead to increased SETBP1 protein abundance. Nevertheless, we have run further experiments and now include a new WB image showing degron versus non-degron variants, as well as quantifying the results (now Fig 2, under Results section “**SETBP1 missense variants outside the degron cause accumulation of SETBP1 protein and increased proliferation in patient fibroblasts independent of SET/PP2A axis**”, lines 367-424). The results of our new analyses confirmed our original conclusion that a subset of germline *SETBP1* variants outside the degron lead to accumulation of SETBP1 protein and promote cell proliferation via a mechanism that is not driven by alterations in SETBP1/SET/PP2A interaction.

3. In Figure 2C, they make the point that exogenously expressed SETBP1 variants (vs. WT) display elevated abundance, which is quantified in Figure 2D. However, it does not appear that

they validated equal expression of the exogenously expressed transcripts to ensure that the variants are simply not more highly transcribed vs. WT (unless I missed this). This should be performed. Also, while it appears true that the conglomerate effect of the variants (vs. WT) do not impact SET interactions or pPP2A/PP2A levels, it seems that some of the variants may indeed affect these phenomena (e.g., E858K vs. control seems to lead to increased pPP2A). Is this just an issue of variability in sample quantifications, or do the authors believe that some of the variants may indeed have differential effects?

A graph showing expression of the exogenously expressed transcripts has been added in Supplementary Fig 4b. Although the variants are more exogenously expressed at the transcript level, we performed extensive analysis on the degradation of the accumulation of SETBP1 proteins by drug inhibition of the protein degradation pathways and showed that the protein degradation of the tested SETBP1 variants was impaired (Fig. 3).

The current Fig 2F in our revised manuscript shows that the pPP2A/PP2A ratios of the tested variants in patient HDFs are slightly lower than controls, but the differences are not statistically significant. Our results are consistent with the results of Banfi et al., 2021¹, who showed that although SET levels in both iPSCs and neural progenitors were significantly different for SGS variants compared to controls, the ratios of pPP2A/PP2A did not differ. Thus, effects on the SETBP1/SET/PP2A axis might indeed be specific to leukaemia cells but not other cell types.

4. The observation that “fibroblasts carrying SETBP1 variants outside the degron showed distinct transcriptomic profiles from healthy control, SGS and SETBP1-haploinsufficiency disorder patients,” while true to some degree, seems overstated. In fact, from the data in Figure 5, it appears that much fewer gene expression differences were observed for variants outside the degron domain vs. within the degron domain, and many of these changes (vs. healthy controls) are similar between the different variants. As such, it seems equally plausible that these overlapping genes might also be important for driving the phenotypic effects in patients. One suggestion might be to perform RRHO analyses to compare the impact of these different variants vs. healthy controls to get a broader sense of how similar/different they may be.

We thank the reviewer for this very helpful suggestion. It is indeed plausible that the overlapping genes might also be important for driving the phenotypic effects in patients, as those would be commonly affected in all SETBP1-related disorders. This is noted in the Discussion of the revised manuscript:

“In our RNA-seq experiments of patient fibroblasts as well as induced neurons derived from patient cells that directly compare the different groups of variants (to our knowledge

the first involving this gene), the three classes of *SETBP1* variants led to distinct transcriptomic profiles affecting different biological pathways with differentially expressed genes that were only partially overlapping. This pattern of findings is further evidence in support of a possible continuum of *SETBP1*-related disorders, with SGS and *SETBP1*-haploinsufficiency disorder positioned at two extremes of severity, involving pathophysiological mechanisms that are partly overlapping along with some features that are unique to each group.” (lines 700-707)

Following the reviewer’s recommendation, we have now performed additional RRHO analyses, which show that the transcriptomic differences among the three groups of variants compared to the controls are different with varying extent of overlaps (Fig 5f and Supplementary Fig 10g).

“To identify overlap between gene-expression signatures of the different variant groups, we used rank–rank hypergeometric overlap (RRHO) analysis⁴. RRHO is a threshold free algorithm that measures the degree of overlap by stepping through two gene lists ranked by the degree of differential expression. These analyses confirmed that DEGs of within-degion variants showed only weak overlap with those of truncating variants. DEGs of outside-degion variants showed moderate overlap with those of truncating variants and only mild overlap with those of within-degion/SGS variants.” (lines 524-536)

We have also revised the Discussion to explicitly address how these results complement the newly added PhenoScore analysis that quantitatively compares the phenotypic similarity of different variant groups. (lines 689-707).

“With the view to aid diagnosis and enhance understanding of the phenotypic spectrum of *SETBP1*-related disorders, we used PhenoScore to quantify similarity of clinical and facial features. Using a model trained with SGS and haploinsufficiency disorder, we further demonstrated that the phenotypic representation associated with variants outside the degion is variable and only partially overlapping with the two previous known disorders (Fig 1). With this approach, we were able to distinguish the heterogenous facial features, a task that was challenging for human eyes. Future studies, including a larger cohort whose clinical and facial features are examined in a standardised way (HPO terms and with photographs taken from different angles), will be beneficial for profiling any dysmorphism, and thus improving deep learning facial feature-based clinical diagnosis.

In our RNA-seq experiments of patient fibroblasts as well as induced neurons derived from patient cells that directly compare the different groups of variants (to our knowledge the

first involving this gene), the three classes of *SETBP1* variants led to distinct transcriptomic profiles affecting different biological pathways with differentially expressed genes that were only partially overlapping. This pattern of findings is further evidence in support of a possible continuum of *SETBP1*-related disorders, with SGS and *SETBP1*-haploinsufficiency disorder positioned at two extremes of severity, involving pathophysiological mechanisms that are partly overlapping along with some features that are unique to each group.” (lines 689-707)

We believe that these additions have significantly increased clarity of the manuscript.

5. Along the lines of #4 above, the authors state that from a transcriptional standpoint that these different variant classes cluster independent (Figure 5B), however, this is overstated, as most of the variant classes appear to cluster together. In addition, there appears to be a tremendous amount of variability among the control samples, so I wonder if the authors could comment on this further.

We are grateful to the reviewer for raising this issue. In our revised manuscript we have modified the statement:

“Fibroblasts carrying different types of *SETBP1* variants formed a continuum, with those outside the degron sitting between within-degron/SGS and truncating variants (Fig 5a).” (lines 498-500)

The control samples were collected from a broad range of unrelated individuals of both biological sexes and different ages, and therefore they are likely to show variability. Three controls were sex-matched parents of the *SETBP1*-haploinsufficiency patients. When performing differential gene expression analysis, we therefore included sex, age, and batch as covariates in the DESeq2 design alongside the variant groups, to regress out the effect of these covariates. In spite of the variability, we were able to detect a number of differentially expressed genes that met statistical significance in the three different variant groups compared to controls, as well as genes overlapping and unique to each group. The findings from the additional RRHO analysis, performed in response to comment 4 above, indicated that the transcriptomic differences showed mild to weak overlaps among different variant groups, i.e. largely different. While DEGs of outside-degron variants are more similar to those of truncating variants, they are relatively less similar to those of within-degron/SGS variants. There was weakest overlap between the DEGs found for within-degron/SGS variants and truncating variants. These findings suggest that outside-degron

variants are more similar to truncating variants but less to within-degron variants while with-degron and truncating variants are the least similar to each other.

In sum, this is an interesting paper that provides much needed insights into the functional consequences of novel SETBP1 variants that result in developmental syndromes. However, numerous issues related to the manner in which the functional data are being interpreted need to be addressed prior to publication.

We thank the reviewer for the positive feedback on our manuscript and the multiple constructive recommendations, and we believe that we have satisfactorily addressed the comments in our revision.

Reviewer #2 (Remarks to the Author):

Wong and colleagues have provided a great description of not only the known SETBP1 variants within the degron or causing haploinsufficiency, but also this novel group of missense variants. In general, the manuscript is well written and easy to follow. I only have a few edits and questions.

We appreciate the reviewer's positive feedback and constructive questions.

1. I am glad you brought up that iPSCs could be used. My first thoughts were "wow this would be great in neural progenitor cells!" But then bringing up that the SGS models were not as helpful as expected is interesting. While certainly cost prohibitive, it is interesting to consider that some of these neurodevelopmental disorder won't work with that model. Are there other disorders that have had similar issues? (of course, they might not be published, so that may be an impossible question!)

We thank the reviewer for suggesting that we should investigate functional impacts of variants in a CNS-relevant model. As also mentioned in our reply to Reviewer #1 point 1, there are thus far three studies that studied the impact of classical SGS variants and *SETBP1* truncating variants in human CNS-relevant models¹⁻³. Results from these studies suggested that defects could already be detected in NPCs and that their proliferation was impaired. The investigation of variant impacts on different subtypes of neurons requires a different study design, long-term culture, and more detailed investigation. This would be a promising follow-up study, and aligns with ongoing efforts in our research group, but is beyond the scope of the current manuscript. Nonetheless, we do recognise the importance of showing that different groups of variants cause defects in neural

cells. This has not been shown in neurons derived from patients carrying *SETBP1* missense variants outside the degron or for LoF variants, and there has not before been any direct comparison across groups of variants. We therefore performed an additional substantial new set of experiments testing the impacts of two variants from each group (classical SGS, missense variants outside degron, and truncating variants) in human induced neurons derived from patient fibroblasts.

The paper has been updated with additional sections in the Methods: “**Generation of induced neurons from fibroblasts using small molecules**” (lines 949-968) and Results: “***SETBP1* variants cause differential effects on differentiation capacity, morphologies, and transcriptomic profiles in induced neurons**” (lines 537-595) and extra Figures showing our findings (Fig 6 and Supplementary Fig 11-14).

We have updated with additional paragraphs in the Discussion that summarised our findings: “**Building on the findings from our broader screen, we showed that the differentiation capacity, morphologies, and transcriptomic profiles of induced neurons derived from patient cells differ from controls, uncovering differential effects for different variant groups, along with variability within groups.**” (lines 632-635)

Moreover, explanations of the choice of neuronal models are added to the Discussion: “**Although patient fibroblasts are not the most severely affected cell type, they are relatively easy to assess and endogenously express *SETBP1*. Since our study placed a strong emphasis on using cell-based functional assays and transcriptomics to screen for pathogenicity and improve definition of *SETBP1*-related disorders, we chose cellular models that allow for systematic functional screening of variants. Investigation of the spatial and temporal expression of *SETBP1* and the functional impact of variants during brain development are much needed to understand the pathways that go awry. This could be achieved by using neuronal models that carry *SETBP1* variants and endogenously express *SETBP1*. To date, only a handful of studies have investigated the impacts of an SGS variant or CRISPR-engineered *SETBP1* deletions or truncating variants in early stage human neural models¹⁻³. However, the impact of *SETBP1* missense or truncating variants have not been studied in neurons and there has not been any direct comparison across groups of variants. Models derived from iPSCs are not suitable for functional screening of variants since the generation of cells carrying selected variants either from patient cells**

or introduced using gene-editing prior to long-term culture of differentiated cells require a different study design, extensive quality control and more detailed investigation, which is labour-intensive. In this study, to systematically compare the neuronal impacts across variant groups, we studied a smaller selection of representative variants using a method that enables relatively rapid generation of induced neurons directly from patient fibroblasts using small molecules. Notably, we showed that the differentiation capacity, morphologies, and transcriptomic profiles of induced neurons derived from patient cells differ from controls, uncovering differential effects for different variant groups, along with variability within groups. We note that the differentiation efficiency and the neuronal subtypes obtained using this method is still relatively limited. Future work comparing the functional impacts of different types of variants in models comprising different neuronal subtypes from different brain regions is essential to further delineate the aetiology, complexity and pleiotropy of SETBP1-related disorders. Importantly, the present study provides valuable insights into which variants should be prioritized for labour-intensive follow-up work involving a subset of variants, chosen based on the results obtained from our functional assays.” (lines 813-842)

We believe that our data comparing effects of missense variants within the degron, outside the degron, and LoF variants, facilitates the understanding of SETBP1-related disorders as a whole in a way that will be important for the field.

2. For your functional studies, should the controls also be age (or approximately age) matched? I'm concerned older individuals may have slower proliferation anyway. You could even use other affected individuals' (just not SETBP1 affected) cells, or try to correct the patient cells with CRISPR (although that's one of the things that might be easier in iPSCs than fibroblasts). Also, a chart (like from GTEx) showing the different isoforms in different cells could be informative.

As also mentioned in response to Reviewer #1 point 1 and your point above, previous studies have suggested impairments in NPCs and neurons derived from iPSCs carrying an SGS or CRISPR-introduced truncating variant. Correcting specific types of variants in patient iPSCs and/or introducing those particular patient variants into control iPSCs, followed by generation of neuronal cells/organoids, would be interesting and valuable as follow-up studies but they require additional much more extensive and elaborate investigations which are far outside the scope of this current manuscript. Already the manuscript contains a very large amount of informative relevant data, especially including the additional induced neuron experiments that we added for

the revision, so we believe that it represents a substantive and comprehensive piece of work in its own right, with many novel findings of interest to the field.

We appreciate the suggestion about showing isoform expression, we have added the recommended information, obtained from the GTEx transcript browser, in Supplementary Fig 1a.

3. I think you touched on this, but is there a clear distinction between the DEGs and the disorders? Looks like Wnt signaling is impaired for variants w/in the degon, could that be partially explanatory of some of the more physical phenotypes observed? You highlight FOXP2, but I think some others could be highlighted as well.

We were able to detect a number of differentially expressed genes in different variant groups compared to controls, as well as genes overlapping and unique to each group. As also mentioned in our response to Reviewer #1 point 4, it is indeed plausible that overlapping genes might also drive the phenotypic effects in patients as those would be commonly affected in all SETBP1-related disorders, in addition to the unique DEGs in each group, which might drive the differences of the groups. As recommended by Reviewer #1, for the revision we performed RRHO analysis of the fibroblast RNA-seq data and found that the global transcriptomic profiles among three groups of variants showed mild to weak overlaps, suggesting that the three variant groups were not similar (Fig 5, Supplementary Fig 10f, lines 524-536). It is indeed possible that differential Wnt signalling contributes to physical phenotypic differences due to its established importance during embryonic development. Importantly, dysregulated Wnt signalling has also been shown in SGS-¹ and *SETBP1*-knockout² NPCs. The original aim of comparing the transcriptomics profiles of fibroblasts was to assess whether we could see differences across different variant groups using a relatively accessible cell type that endogenously expresses SETBP1, with a view to systematic screening. We focused less on finding dysregulated pathways in these cells since they are not the main cell types that are affected in SETBP1-related disorders. Yet, from the results of our new additional experiments run using human induced neurons, we have also been able to show in our revision that different variant groups have differential effects on neuronal differentiation, morphologies, and transcriptomic profiles.

4. You suggest that clinicians use available tools to assess variants. Do you have a suggestion for a protocol for this? You suggest the ion channels tool, which is great, but I think if you could say pathogenic variants have greater than X CADD score or some other tools, that would be great. You have a lot of tools (CADD, SIFT, PolyPhen, MetaDome) but you could also add MTR.

Currently, there is not a consensus on the most appropriate cut-offs of CADD score or other pathogenicity tools to best predict variant pathogenicity. By themselves, these scores are not enough to confirm pathogenicity but can be used as an initial indication of whether a variant is likely to affect protein functions. It is still therefore important to perform functional testing on variants in order to validate pathogenicity. In our study, we used rather stringent cut-off scores for initial screening (e.g. CADD-PHRED scores above 21 etc.), and proceeded with functional assays using different cellular models. As recommended by the reviewer, we have added MTR scores in the tool list (results related to MTR scores are in Supplementary Fig 2d). It is highlighted in the Discussion section:

“...we recommend that clinicians should examine the clinical phenotypes of family members carrying the same variants as well as performing cellular assays to test pathogenicity when assessing pathogenicity of inherited variants of uncertain significance based on ACMG (American College of Medical Genetics and Genomics) guidelines⁵. Clinicians could also consider performing functional prediction using algorithms where available, such as those for known epilepsy ion channel genes⁶.” (lines 678-684)

“Overall, based on our results, when interpreting pathogenicity of variants of unknown significance, we recommend using a wide range of prediction tools and performing functional assays. A list of prediction tools used in our study can be found in section “For More Information”.” (lines 715-718)

5. Are there any SETBP1 duplications? I wonder if that increased protein would be problematic or if the increased protein plus a variant is needed?

Many thanks to the reviewer for this interesting suggestion. We have not found any benign *SETBP1* duplications in ClinGen Benign, ClinVar, or Decipher. In Decipher, we have not found any patients with duplications that only span *SETBP1* without affecting neighbouring genes. There is one patient with a 989.55 kb duplication that has global developmental delay and intellectual disability, but the duplicated region includes both *SETBP1* and *SCL14A2* and therefore it is unclear whether a duplication of *SETBP1* alone can be pathogenic. In addition, it is hard to conclude whether a *SETBP1* duplication would be enough to cause a disorder with phenotypic profiles similar to SGS or haploinsufficiency disorder because it depends on multiple factors such as the genomic location/arrangement of the duplication, including whether it disrupts *SETBP1* (yielding haploinsufficiency) and/or whether it occurs as a tandem duplication (potentially causing increased expression of the gene). SGS variants are thought to cause high SETBP1 protein

abundance, but it is still plausible that such variants act via mechanisms other than merely increasing protein abundance, an issue which remains unclear for the field.

6. Finally, I think there is an opportunity for a helpful figure, something that connects all 3 SETBP1-related disorders. Maybe a spectrum along the protein or a chart? There is a lot of information on the three disorders, so having a figure and/or table would be helpful. Kind of like Figure 6, but for phenotypes or HPO terms.

We agree that it would be helpful to summarise the clinical data in a way that makes the paper more accessible to readers. In the revised manuscript, we added new analyses using PhenoScore⁷, an artificial intelligence-based phenomics framework that combines state-of-the-art facial recognition technology using patient photographs with analysis of phenotypic data in Human Phenotype Ontology (HPO) terms to quantify phenotypic similarity. The results obtained provide a useful overview of the SETBP1-related disorders investigated in our study, and are now included in Fig. 1c-d and Supplementary Table 2, as well as in the Results section:

“Next we sought to use a quantitative method to better understand the phenotypic differences of individuals with *SETBP1* variants and to aid diagnosis. We therefore utilized PhenoScore⁷ to quantitatively investigate whether individuals included in this study were more similar to individuals with *SGS* or to individuals with *SETBP1*-haploinsufficiency disorder. PhenoScore is an artificial intelligence-based phenomics framework that combines state-of-the-art facial recognition technology with analysis of phenotypic data in Human Phenotype Ontology (HPO) terms to quantify phenotypic similarity. We first performed a subgroup analysis to demonstrate that PhenoScore was able to distinguish *SGS* and *SETBP1*-haploinsufficiency disorder (phenotypic data and facial photographs of five individuals from each group) (Fig 1c). We then generated individual predictions for all individuals with a missense variant in *SETBP1* outside the degron included in this study using this trained PhenoScore model, to determine whether these were more phenotypically similar to the individuals with *SETBP1*-haploinsufficiency disorder (score = 0) or with *SGS* (score = 1). We performed this prediction for facial features and HPO terms separately, and also as a combined prediction (PhenoScore). Intriguingly, when only considering HPO terms alone, the majority of the individuals with variants outside the degron were more similar to those with haploinsufficiency disorder, or did not match with either *SGS* or haploinsufficiency disorder. However, the facial features of those with a variant directly adjacent to the degron were more similar to *SGS* while those further away from degron were more similar to haploinsufficiency disorder, suggesting a gradient for

craniofacial feature formation. Interestingly, none of the tested variants outside the degron showed a PhenoScore similar to SGS; instead the majority were either classified as no match, while a few were classified as haploinsufficiency disorder (Fig 1d, Supplementary Fig 1b-c, Supplementary Table 2). Thus, the clinical presentations of variants outside the degron were distinct from classical SGS and sometimes even from truncating variants, as could be distinguished using quantitative phenotypic features.” (lines 260 - 285)

A few edits:

Abstract:

Line 70-

This sentence in the abstract is too long and hard to read: “However, due to the lack of systematic investigation of genotype-phenotype associations of different types of SETBP1 variants, and limited understanding of its roles in neurodevelopment, the extent of clinical heterogeneity and how this relates to underlying pathophysiological mechanisms remain elusive, imposing challenges for diagnosis.”

We modified the sentence:

“However, due to the lack of systematic investigation of genotype-phenotype associations of different types of SETBP1 variants, and limited understanding of its roles in neurodevelopment, the extent of clinical heterogeneity and how this relates to underlying pathophysiological mechanisms remain elusive. This imposes challenges for diagnosis.”
(lines 72-74)

Line 75-

I would say “degron region” since you haven’t had a chance to define it yet.

We modified the sentence as recommended:

“Here, we present a comprehensive investigation of the largest cohort to-date of individuals carrying SETBP1 missense variants outside the degron region (n=18).” (lines 76-77)

Results:

Line 208-

What is MSH?

MSH are the initials of Michael S Hildebrand, one of the co-authors of the manuscript. We modified the sentence to make it clear:

“However, these variants in probands 1 and 3 are unlikely to be pathogenic based on the patients’ clinical features, the inheritance model, and the lack of functional impact observed in assays performed in cellular models (unpublished observations by co-author M.S.H. with the *EHMT1* and *WWOX* variants).” (lines 213-216)

Line 220-

I think midface retrusion is more commonly used

We modified the sentence as recommended.

“Shared facial features were present in at least three out of the five individuals, including prominent ears, shallow orbits, midface retrusion and microcephaly (Fig 1Bb).” (lines 226-228)

Line 225 –

Move the “(amino acids 854-858)” to after “within the SKI domain...”

We modified the sentence as recommended:

“Individuals (n=7) with variants located slightly further away from the degron but still within the SKI domain (amino acids 854-858) showed mild or moderate intellectual disability.” (lines 232-233)

Line 310-

This is explaining your strategy, but does not give the results, maybe change the formatting?

We moved the paragraph to Section **“*SETBP1* missense variants outside the degron cause accumulation of *SETBP1* protein and increased proliferation in patient fibroblasts independent of *SET/PP2A* axis”** before showing the results of the functional assays. (lines 344-358)

Overall, this is a really nice paper explaining a complicated set of disorders!

We thank the reviewer for the positive feedback and constructive comments on our manuscript.

Reviewer #3 (Remarks to the Author):

The study conducted by Wong and colleagues aims to illustrate the heterogeneity in molecular and clinical characteristics of *SETBP1* variants through the use of in vitro functional assays and

transcriptomic analyses. Interestingly, the authors could show that patients' phenotypical expression varies and seems to be influenced by the proximity to the degron region. Therefore, the authors selected and tested various variants in their subsequent experiments, differentiating and considering their distances between those located within and outside of the canonical degron.

Wong and colleagues show that a subset of SETBP1 variants outside the degron leads to an increase in protein levels and promotes cell proliferation. Moreover, SETBP1 degradation via autophagy or proteasomal pathways was impacted, but the degree and direction of degradation varied depending on the variant's distance from the degron region. Similarly, a subset of variants showed reduced transcriptional activity.

Lastly, the authors performed RNA-seq in patient-derived fibroblasts, showing distinct differential expression patterns depending on SETBP1 variants.

The study represents a valuable resource for clinical diagnostic and mechanistic research. In addition, the study represents a very nice example of the degree of heterogeneity of phenotypes, even when different variants are studied side by side. To this extent, Figure 6 is very impactful.

We appreciate the reviewer's positive comments on our manuscript.

However, several aspects limit the impact of this study and should be considered to improve the manuscript.

1) The role of the observed phenotypes DNA-binding/transcription in the emergence of the clinical presentation remains elusive. This is a critical limitation of the manuscript. The authors should either develop this aspect further (see also comments on the cell type used) or remove some (over)statements and highlight this limitation throughout the manuscript.

We thank the reviewer for this comment. The aim of DNA-binding/transcription assays was to establish a systematic way to screen in parallel multiple different variants for pathogenicity in cellular models. We then assessed the impacts of a selected subset of variants on the overall transcriptomic profiles of patient fibroblasts to further establish that there are effects on global gene expression. We acknowledge that in the initial manuscript, experiments were performed only in transfected cell lines and patient fibroblasts. As we also explained in our responses to previous reviewers' comments, the original aim of comparing the transcriptomics profiles of fibroblasts was to assess whether we could see differences among different variant groups, using a relatively accessible cell type that endogenously expresses SETBP1, making it suitable for

screening purposes. We focused less on finding dysregulated pathways in these cells since they are not the main cell types that are affected in SETBP1-related disorders. Importantly, from the new experiments with human induced neurons, we observed that different variant groups had differential effects on neuronal differentiation, morphologies, and transcriptomic profiles.

Moreover, for our revised manuscript, we have explicitly highlighted in the Discussion both the advantages and limitations of the methods used in this study:

“Although patient fibroblasts are not the most severely affected cell type, they are relatively easy to assess and endogenously express SETBP1. Since our study placed a strong emphasis on using cell-based functional assays and transcriptomics to screen for pathogenicity and improve definition of SETBP1-related disorders, we chose cellular models that allow for systematic functional screening of variants. Investigation of the spatial and temporal expression of SETBP1 and the functional impact of variants during brain development are much needed to understand the pathways that go awry. This could be achieved by using neuronal models that carry *SETBP1* variants and endogenously express SETBP1. To date, only a handful of studies have investigated the impacts of an SGS variant or CRISPR-engineered *SETBP1* deletions or truncating variants in early stage human neural models¹⁻³. However, the impact of *SETBP1* missense or truncating variants have not been studied in neurons and there has not been any direct comparison across groups of variants. Models derived from iPSCs are not suitable for functional screening of variants since the generation of cells carrying selected variants either from patient cells or introduced using gene-editing prior to long-term culture of differentiated cells require a different study design, extensive quality control and more detailed investigation, which is labour-intensive. In this study, to systematically compare the neuronal impacts across variant groups, we studied a smaller selection of representative variants using a method that enables relatively rapid generation of induced neurons directly from patient fibroblasts using small molecules. Notably, we showed that the differentiation capacity, morphologies, and transcriptomic profiles of induced neurons derived from patient cells differ from controls, uncovering differential effects for different variant groups, along with variability within groups. We note that the differentiation efficiency and the neuronal subtypes obtained using this method is still relatively limited. Future work comparing the functional impacts of different types of variants in models comprising different neuronal subtypes from different brain regions is essential to further delineate the aetiology, complexity and pleiotropy of SETBP1-related disorders. Importantly, the present study

provides valuable insights into which variants should be prioritized for labour-intensive follow-up work involving a subset of variants, chosen based on the results obtained from our functional assays.” (lines 813-842)

2) The authors claim that they have identified FOXP2 as a direct transcriptional target of SETBP1 through their cell-based experiments. However, they have not provided any direct evidence to support this conclusion, leaving the reasoning behind it unclear. In line with that, the results depicted in Figure 4C show reporter constructs for FOXP2 promoters without giving a rationale for choosing this transcriptional target. Furthermore, the authors generated RNA-seq data from multiple patient fibroblasts that could be utilized to confirm the transcriptional regulation of their target FOXP2. Is there any indication within the RNA-seq dataset that suggests FOXP2 is differentially regulated in mutant cells? The authors should comment on these findings.

We thank the reviewer for pointing this out. *FOXP2* was originally identified as a putative direct target of SETBP1 in a ChIP-seq dataset of HEK cells expressing tagged SETBP1⁸ (Fig. S3 in Piazza et al., 2018). Inspired by this, we therefore validated this interaction using luciferase-reporter assays and compared the impacts of different missense variants. In our revision, we have modified the relevant parts of the text to make this clearer (lines 471-476 of revised manuscript): **“A previously published chromatin immunoprecipitation sequencing (ChIP-Seq) dataset showed that *FOXP2*, rare genetic disruptions of which lead to CAS⁹⁻¹¹, was one of the 70 putative SETBP1 targets in HEK cells expressing a wild type SETBP1 construct⁸. We therefore first validated *FOXP2* as a novel direct transcriptional target of SETBP1 and demonstrated that it could be activated by *SETBP1* at two different sites using a luciferase reporter assay (Fig 4c).”**

FOXP2 was not identified as a DEG in our RNA-seq experiment using patient fibroblasts, which could reflect cell type-specific effects or might alternatively relate to the low expression of both *SETBP1* and *FOXP2* in fibroblasts. The putative regulation of FOXP2 by SETBP1 was recently confirmed in analyses of zebra finch orthologs of these genes, as reported by our group and colleagues¹². Notably, *FOXP2* is a significant DEG in the induced neurons in our current study, further validating the findings, and indicating that the effects could indeed be cell type-specific. We have added commentary on this in our revised manuscript (lines 730-735):

“Notably, *FOXP2* was a significant DEG in our patient cell-derived induced neurons but not in patient fibroblasts, or in those performed on patient leukaemia cells and cell lines expressing *SETBP1* constructs in previous studies, further pointing towards potential cell

type-specific aetiological pathways, Moreover, we have recently demonstrated that the avian ortholog of *SETBP1* regulates *FoxP2* in the zebra finch, which is a well-established model of vocal learning, using a luciferase reporter assay¹².”

3) According to the authors, variants located farthest from the degron (specifically, p.(Ser444Arg) and p.(Val657Ala)) exhibit impaired SETBP1 degradation due to disrupted ubiquitination. However, the data presented in Figure 3 only partially support these findings, as the ubiquitination levels of both variants are similar to those of wild-type SETBP1. To avoid misinterpretation of results, the western blot (Fig. 3C) should be repeated since the input of wild-type SETBP1 appears to be at lower levels compared to the other variants, as indicated by the reduced intensity of the loading control β -actin.

The level was normalised to the amount of input SETBP1, and thus the proportion of SETBP1 that is ubiquitinated is lower regardless of the absolute level of SETBP1. There is higher expression of the mutated SETBP1 protein due to the presence of the variant and that is why there is more mutated protein. We therefore did not repeat the experiment, but text is included in the Methods to clarify this point.

“For quantification of (co-) immunoprecipitation experiments, the background-subtracted band intensity of the immunoprecipitated fraction was normalised with respect to the input fraction for each condition¹³.” (lines 1136-1140)

4) The authors have emphasized at different points that the causes and outcomes related to SETBP1 variants are likely to be specific to particular cell types. However, the current manuscript employs different cell types. Hence, it is necessary for the authors to address how this may affect their results.

We thank the reviewer for pointing this out. We acknowledge that in the initial manuscript, experiments were performed only in transfected cell lines and patient fibroblasts. As we also explained in our responses to previous reviewers' comments and point 1 above, the original aim of comparing the transcriptomics profiles of fibroblasts was to assess whether we could see differences among different variant groups, using a relatively accessible cell type that endogenously expresses SETBP1, making it suitable for screening purposes. We focused less on finding dysregulated pathways in these cells since they are not the main cell types that are affected in SETBP1-related disorders. There are thus far three studies that studied the impact of classical SGS variants, *SETBP1*-knockout and CRI in human CNS-relevant models¹⁻³. Results from these studies suggested that defects could already be detected in NPCs and that their

proliferation was impaired. The investigation of variant impacts on different subtypes of neurons requires a different study design, long-term culture, and more detailed investigation. This would be a promising follow-up study, and aligns with ongoing efforts in our research group, but is beyond the scope of the current manuscript. Nonetheless, we do recognise the importance of showing that different groups of variants cause defects in neural cells. This has not been shown in neurons derived from patients carrying *SETBP1* missense variants outside the degron or for LoF variants, and there has not before been any direct comparison across groups of variants. We therefore performed an additional substantial new set of experiments testing the impacts of two variants from each group (classical SGS, missense variants outside degron and truncating variants) in human induced neurons derived from patient fibroblasts. Importantly, from the new experiments with human induced neurons, we observed that different variant groups had differential effects on neuronal differentiation, morphologies, and transcriptomic profiles.

The paper has been updated with additional sections in the Methods: “**Generation of induced neurons from fibroblasts using small molecules**” (lines 949-968) and Results: “***SETBP1* variants cause differential effects on differentiation capacity, morphologies, and transcriptomic profiles in induced neurons**” (lines 537-595) and extra Figures showing our findings (Fig 6 and Supplementary Fig 11-14).

We have updated with additional paragraphs in the Discussion that summarised our findings: “**Building on the findings from our broader screen, we showed that the differentiation capacity, morphologies, and transcriptomic profiles of induced neurons derived from patient cells differ from controls, uncovering differential effects for different variant groups, along with variability within groups.**” (lines 632-635)

Moreover, for our revised manuscript, we have explicitly highlighted in the Discussion both the advantages and limitations of the methods used in this study:

“**Although patient fibroblasts are not the most severely affected cell type, they are relatively easy to assess and endogenously express *SETBP1*. Since our study placed a strong emphasis on using cell-based functional assays and transcriptomics to screen for pathogenicity and improve definition of *SETBP1*-related disorders, we chose cellular models that allow for systematic functional screening of variants. Investigation of the spatial and temporal expression of *SETBP1* and the functional impact of variants during brain development are much needed to understand the pathways that go awry. This could**

be achieved by using neuronal models that carry *SETBP1* variants and endogenously express SETBP1. To date, only a handful of studies have investigated the impacts of an SGS variant or CRISPR-engineered *SETBP1* deletions or truncating variants in early stage human neural models¹⁻³. However, the impact of *SETBP1* missense or truncating variants have not been studied in neurons and there has not been any direct comparison across groups of variants. Models derived from iPSCs are not suitable for functional screening of variants since the generation of cells carrying selected variants either from patient cells or introduced using gene-editing prior to long-term culture of differentiated cells require a different study design, extensive quality control and more detailed investigation, which is labour-intensive. In this study, to systematically compare the neuronal impacts across variant groups, we studied a smaller selection of representative variants using a method that enables relatively rapid generation of induced neurons directly from patient fibroblasts using small molecules. Notably, we showed that the differentiation capacity, morphologies, and transcriptomic profiles of induced neurons derived from patient cells differ from controls, uncovering differential effects for different variant groups, along with variability within groups. We note that the differentiation efficiency and the neuronal subtypes obtained using this method is still relatively limited. Future work comparing the functional impacts of different types of variants in models comprising different neuronal subtypes from different brain regions is essential to further delineate the aetiology, complexity and pleiotropy of SETBP1-related disorders. Importantly, the present study provides valuable insights into which variants should be prioritized for labour-intensive follow-up work involving a subset of variants, chosen based on the results obtained from our functional assays.” (lines 813-842)

We believe that our data comparing effects of missense variants within the degron, outside the degron, and LoF variants, facilitates the understanding of SETBP1-related disorders as a whole in a way that will be important for the field.

5) Related to the point above, the authors in several points claim that their work addresses the pathophysiological mechanisms of SETDP1 mutations. However, as patients with mutations in this gene have CNS issues, more work on CNS-relevant cells/models would better support this claim. This is one of the major limitations of this study.

We are grateful to the reviewer for this comment. As also elaborated on in our responses to reviewer #1 point 4, reviewers #2 point 1, and points 1 and 4 above, in the revised manuscript we provide an entire new series of experiments investigating the impacts of two variants from each

group (classical SGS, missense variants outside degron, and truncating variants) in human induced neurons derived from patient fibroblasts. Importantly, from the new experiments with human induced neurons, we observed that different variant groups had differential effects on neuronal differentiation, morphologies, and transcriptomic profiles.

The paper has been updated with additional sections in the Methods: **“Generation of induced neurons from fibroblasts using small molecules”** (lines 949-968) and Results: **“SETBP1 variants cause differential effects on differentiation capacity, morphologies, and transcriptomic profiles in induced neurons”** (lines 537-595) and extra Figures showing our findings (Fig 6 and Supplementary Fig 11-14).

We have updated with additional paragraphs in the Discussion that summarised our findings: **“Building on the findings from our broader screen, we showed that the differentiation capacity, morphologies, and transcriptomic profiles of induced neurons derived from patient cells differ from controls, uncovering differential effects for different variant groups, along with variability within groups.”** (lines 632-635)

Moreover, for our revised manuscript, we have explicitly highlighted in the Discussion both the advantages and limitations of the methods used in this study (lines 813-842).

We believe that our data comparing effects of missense variants within the degron, outside the degron, and LoF variants, facilitates the understanding of SETBP1-related disorders as a whole in a way that will be important for the field.

6) According to Wong et al., SETBP1 variants impact multiple protein functions. While functional assays and western blots are valuable for validating their hypotheses, proteomic analyses would have been a more suitable option. This is particularly relevant since the authors noted differences in protein degradation, which could have allowed for the identification of direct consequences on protein levels and potential interactions with SETBP1. Additionally, these findings could have been cross-referenced with RNA-seq data to explore direct SETBP1 interactors.

We thank the reviewer for this suggestion. It will indeed be valuable to study the interaction of SETBP1 mutants with other protein interactors using proteomics in future studies, but these are beyond the scope of the current work, which already encompasses a large substantial amount of

information covering an array of models and analytical levels. In our revised manuscript we have added commentary about the value of proteomics strategies to the Discussion:

“Future studies that profile and delineate the structural impact of *SETBP1* variants and how they affect interactions with cofactors, for example via proteomic approaches, will aid the understanding of their impacts on protein functions and thus aetiology.” (lines 789-799)

7) The authors investigate transcriptomic profiles from different *SETBP1* variants, pointing out that the various modalities (outside, within, truncating) are largely separated clusters (ad Fig. 6B). However, as indicated in Fig 6B, the variants are actually intermingled, showing an overlap throughout. The authors should revise and comment on that.

We thank the reviewer for pointing this out, which is also in line with Reviewer # 1 point 4. We have now revised the sentence.

“Fibroblasts carrying different types of *SETBP1* variants formed a continuum, with those outside the degron sitting between within-degron/SGS and truncating variants (Fig 5a).” (lines 498-500)

Following Reviewer #1’s recommendation, we have now performed additional RRHO analyses, which show that the transcriptomic differences among the three groups of variants compared to the controls are different with varying extent of overlaps (Fig 5f and Supplementary Fig 10g).

“To identify overlap between gene-expression signatures of the different variant groups, we used rank–rank hypergeometric overlap (RRHO) analysis⁴. RRHO is a threshold free algorithm that measures the degree of overlap by stepping through two gene lists ranked by the degree of differential expression. These analyses confirmed that DEGs of within-degron variants showed only weak overlap with those of truncating variants. DEGs of outside-degron variants showed moderate overlap with those of truncating variants and only mild overlap with those of within-degron/SGS variants” (lines 524-536)

We have also revised the Discussion to explicitly address how these results complement the newly added PhenoScore analysis that quantitatively compares the phenotypic similarity of different variant groups.

“With the view to aid diagnosis and enhance understanding of the phenotypic spectrum of *SETBP1*-related disorders, we used PhenoScore to quantify similarity of clinical and facial features. Using a model trained with SGS and haploinsufficiency disorder, we further

demonstrated that the phenotypic representation associated with variants outside the degron is variable and only partially overlapping with the two previous known disorders (Fig 1). With this approach, we were able to distinguish the heterogenous facial features, a task that was challenging for human eyes. Future studies, including a larger cohort whose clinical and facial features are examined in a standardised way (HPO terms and with photographs taken from different angles), will be beneficial for profiling any dysmorphism, and thus improving deep learning facial feature-based clinical diagnosis.

In our RNA-seq experiments of patient fibroblasts as well as induced neurons derived from patient cells that directly compare the different groups of variants (to our knowledge the first involving this gene), the three classes of *SETBP1* variants led to distinct transcriptomic profiles affecting different biological pathways with differentially expressed genes that were only partially overlapping. This pattern of findings is further evidence in support of a possible continuum of SETBP1-related disorders, with SGS and *SETBP1*-haploinsufficiency disorder positioned at two extremes of severity, involving pathophysiological mechanisms that are partly overlapping along with some features that are unique to each group (lines 689-707).”

We believe that these additions have significantly increased clarity of the manuscript.

Minor comments:

1) The figure panels are relatively compact, making it challenging to differentiate between individual data markers, particularly in Fig. 2 and 3.

We have increased the size and clarity of the panels in these Figures.

2) The labeling in Figure 3C,D is incorrect as the variant p.(Val657Ala) is erroneously abbreviated as "V647A". The authors should correct this error.

We have corrected this error.

3) Ad line 371: “to inhibit” not “to inhibition”

We have corrected this (“to inhibit”, now line 402).

4) ad line 691: The sentence contains a fraction written in italics that should be changed to a regular font.

We made the change (“a subset of genes”, now line 802).

5) ad line 707: The sentence abruptly concludes with the phrase "in SETBP1-related disorders and. To study..."; this part needs to be revised to improve clarity and comprehension.

We modified the paragraph to improve clarity.

References

1. Banfi, F. *et al.* SETBP1 accumulation induces P53 inhibition and genotoxic stress in neural progenitors underlying neurodegeneration in Schinzel-Giedion syndrome. *Nat Commun* **12**, 4050 (2021).
2. Cardo, L. F., de la Fuente, D. C. & Li, M. Impaired neurogenesis and neural progenitor fate choice in a human stem cell model of SETBP1 disorder. *Molecular Autism* **14**, 8 (2023).
3. Shaw, N. C. *et al.* Identifying SETBP1 haploinsufficiency molecular pathways to improve patient diagnosis using induced pluripotent stem cells and neural disease modelling. *Molecular Autism* **15**, 42 (2024).
4. RRHO. *Bioconductor* <http://bioconductor.org/packages/RRHO/>.
5. Richards, S. *et al.* Standards and guidelines for the interpretation of sequence variants: a joint consensus recommendation of the American College of Medical Genetics and Genomics and the Association for Molecular Pathology. *Genet Med* **17**, 405–423 (2015).
6. Heyne, H. O. *et al.* Predicting functional effects of missense variants in voltage-gated sodium and calcium channels. *Science Translational Medicine* **12**, eaay 8648 (2020).
7. Dingemans, A. J. M. *et al.* PhenoScore quantifies phenotypic variation for rare genetic diseases by combining facial analysis with other clinical features using a machine-learning framework. *Nat Genet* **55**, 1598–1607 (2023).
8. Piazza, R. *et al.* SETBP1 induces transcription of a network of development genes by acting as an epigenetic hub. *Nat Commun* **9**, 2192 (2018).
9. Lai, C. S. L., Fisher, S. E., Hurst, J. A., Vargha-Khadem, F. & Monaco, A. P. A forkhead-domain gene is mutated in a severe speech and language disorder. *Nature* **413**, 519–523 (2001).

10. Reuter, M. S. *et al.* FOXP2 variants in 14 individuals with developmental speech and language disorders broaden the mutational and clinical spectrum. *Journal of Medical Genetics* **54**, 64–72 (2017).
11. Morgan, A., Fisher, S. E., Scheffer, I. & Hildebrand, M. FOXP2-Related Speech and Language Disorders. in *GeneReviews®* (eds. Adam, M. P. et al.) (University of Washington, Seattle, Seattle (WA), 2016).
12. Grönberg, D. *et al.* Expression and regulation of SETBP1 in the song system of male zebra finches (*Taeniopygia guttata*) during singing. *bioRxiv* 2024.06.05.597622 (2024) doi:10.1101/2024.06.05.597622.
13. Burckhardt, C. J., Minna, J. D. & Danuser, G. Co-immunoprecipitation and semi-quantitative immunoblotting for the analysis of protein-protein interactions. *STAR Protocols* **2**, 100644 (2021).

Response to reviewer comments

We would like to thank the reviewers for their positive feedback. Our detailed point-by-point responses to each reviewer comment are given below in blue. Quoted newly added text from our main manuscript are “**in blue + bold**”.

Reviewer #1 (Remarks to the Author):

In this substantially revised manuscript, the authors have done a commendable job in responding to my previous critiques, as well as those of my fellow Reviewers, both through the addition of new data (including from iPSC neurons, as requested) and text revisions. I now feel that this manuscript is suitable for publication at Nature Communications and will be of broad interest to the field.

We are grateful to the reviewer for this positive assessment and for recognizing the value of our integrated approach and additional experiments to bring together clinical phenotyping and cell-based functional assays.

Reviewer #2 (Remarks to the Author):

The authors nicely describe SETBP1-related disorders and how they vary by variant type and location, particularly the missense variants outside the degron region. Having multiple variables (variant type, location, and phenotype) leads to complex analyses. The clinical phenotyping and molecular studies support their conclusions well. These findings expand the known features of SETBP1-related disorders and provide molecular studies to be used to identify therapeutics.

This work is very significant to the SETBP1-related disorders field. Additionally, their approach will likely benefit other neurodevelopmental disorders with varying missense variants and hotspots. They also use an interesting approach with correlating the phenotypes, facial features, and variant location that I think will be relevant to other disorders.

The in depth molecular phenotyping provides a wealth of novel information. Being able to compare the different variant types across multiple cell types is very valuable. This is a very nice manuscript that provides very important information.

We thank the reviewer for the positive feedback and for recognizing the value of our integrated approach to bring together clinical phenotyping and cell-based functional assays.

Reviewer #3 (Remarks to the Author):

The authors addressed all my criticisms and revised the manuscript to recognize the limitations in the experimental design and model systems used. The overall organization of the manuscript has improved. The authors have now utilized induced neurons for transcriptomics, significantly improving the quality and insights regarding SETBP1 variants.

We appreciate the reviewer’s positive comments on our manuscript and recognizing the values of our additional experiments and analyses.

I suggest thoroughly checking the language and figure references, as some mistakes remain. Below are some examples I have noted.

We have revised the text and figure references of the manuscript in addition to the following changes as suggested by the Reviewer.

1. Line 274: „...and to aid diagnosis.”
We have corrected this to “and to aid diagnosis”, now line 257.
2. Line 394: refer to Suppl. Fig. 4a not b
We have corrected this to “Supplementary Fig 4a”, now line 358.
3. Line 399: refer to Suppl. Fig. 4b not c
We have corrected this to “Supplementary Fig 4b”, now line 363.
4. Line 405: refer to Suppl. Fig. 4c (else there is no reference to this panel in the whole text)
We have corrected this to “Supplementary Fig 4c-f”, now line 371.
5. Line 421: there is no Suppl. Fig. 4i
We have corrected this to “Supplementary Fig 4g-h”, now line 383.
6. Line 427: there is no Suppl. Fig. 4j
We have corrected this textual mistake, the text “Supplementary Fig 4j” was deleted, now line 396.
7. Line 563: refer to Suppl. Fig. 10b-f
We have corrected this error; this part was deleted.
8. Line 609: (“days 10 and12”) put a space before 12
We have corrected this to “days 10 and 12”, now line 548.
9. Line 939: there is no Suppl. Figure 5d
We have corrected this textual mistake, the text “Supplementary 5d” was deleted, now line 848.